# CAUSAL CARTOGRAPHER: FROM MAPPING TO REASONING OVER COUNTERFACTUAL WORLDS

## ABSTRACT

Causal world models are systems that can answer counterfactual questions about an environment of interest, i.e., predict how it would have evolved if an arbitrary subset of events had been realized differently. The ability to answer such questions is crucial for models to reliably understand the world. However, this task currently eludes large language models (LLMs), which do not have demonstrated causal reasoning capabilities beyond the memorization of existing causal relationships. Furthermore, evaluating counterfactuals in real-world applications is challenging since only the factual world is observed, limiting evaluation to synthetic datasets. We address these problems by proposing the Causal Cartographer, a twofold system composed of two agents: the first extracts causal relationships from data and builds a vast repository of causal knowledge, while the second uses them as constraints to perform reliable step-by-step causal inference. We evaluate our approach on real-world counterfactuals obtained by matching data from diverse news sources. We show that our approach can extract accurate causal knowledge and enhance the robustness of LLMs for causal reasoning tasks. In particular, the proposed causal conditioning mitigates the impact of spurious correlations and greatly reduces inference costs (by up to 70%) compared to chain-of-thought reasoning.

## 1 INTRODUCTION

Learning to infer causal relationships and making causal predictions about the world is an important task for general reasoning systems (Goyal & Bengio, 2020; Schölkopf et al., 2021). In particular, predicting how an environment would have evolved under a different policy (i.e. counterfactual questions) is a challenging and crucial question when evaluating how an artificial system understands the world. While an agent with low causal knowledge can make predictions about observed distributions, generalizing to arbitrary distributions and counterfactuals requires one to learn a causal world model (Bareinboim et al., 2022; Richens & Everitt, 2024). Studies on large language models (LLMs) have shown that they do not perform robust causal discovery or causal inference (Zecevic et al., 2023; Jin et al., 2023; 2024; Chen et al., 2024; Jiralerspong et al., 2024; Joshi et al., 2024) and fail to generalize to unseen distributions (Wu et al., 2024; Gendron et al., 2024a;b; Berglund et al., 2024). This is a challenging task as causal knowledge is notoriously hard to collect (Rubin, 1974; Pearl, 2009) and counterfactual data is generally not available as only the factual world is observed (Holland, 1986).

We tackle these problems by explicitly modeling causal relationships and enhancing LLM agents with a causal reasoning framework. We introduce the **Causal Cartographer** [1], illustrated in Figure 1: a twofold system composed of a *graph retrieval-augmented generation* (Graph-RAG (Lewis et al., 2020; Edge et al., 2024; Peng et al., 2024)) *agent* tasked to retrieve causal relationships from real-world news articles and a *counterfactual reasoning agent* performing reliable and efficient step-by-step causal inference while respecting causal relationships. We take advantage of the first component to build CausalWorld, a causal network that maps causal knowledge of the world, which we use to *provably* access counterfactual knowledge and build real-world causal questions for our second reasoning agent. We use the extracted knowledge to guide our causal reasoning agent on real-world counterfactual reasoning tasks. At every reasoning step, the context of the agent is restricted to information from the causal parents, reducing context length while eliminating spurious correlations.

---

[1] Our code is available as an anonymous repository at: `https://anonymous.4open.science/r/causal-world-modelling-agent-CECF`

We show that our proposed method accurately extracts causal knowledge from natural language data and allows estimating real-world counterfactual situations. Our contributions are as follows:

- We introduce the Causal cartographer: a twofold system composed of (1) a causal extraction method for unstructured natural language data with a graph retrieval-augmented generation (Graph-RAG) agent, and (2) a step-by-step causal inference agent that can perform counterfactual reasoning while respecting causal constraints;

- We use our extraction method to build CausalWorld, a network of causal relationships from real-world news events published in 2020, containing 975 variables and 1337 causal relationships, and use this network as a repository of causal knowledge to evaluate large language models on counterfactual reasoning with real-world natural language data;

- We introduce the theoretical notion of causal blankets and prove that, assuming identifiability of the blanket, we can sample true counterfactuals from a causal network;

- We evaluate our causal inference agent and show that causal conditioning achieves competitive performance for counterfactual reasoning while being more robust and greatly reducing the LLM's context window and output length, decreasing the inference cost up to 70%.

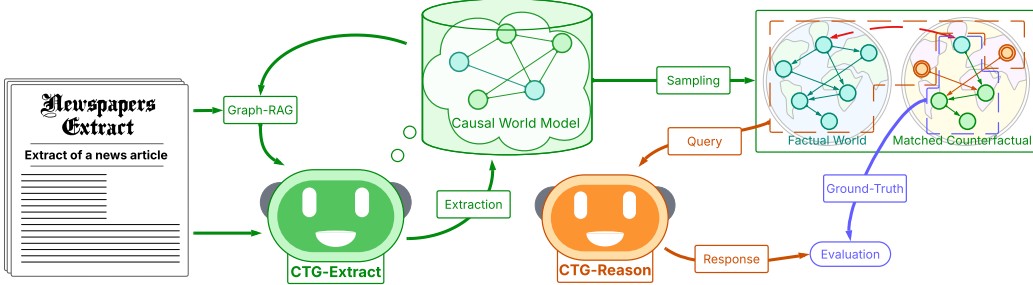

Figure 1: Overview of the Causal Cartographer. The title, content and metadata of a news source are provided to the extraction agent (CTG-Extract). It generates causal variables and their relationships and adds them to the causal world model, making causal knowledge explicit. A Graph-RAG system further guides the extraction process. Then, ground-truth counterfactuals are sampled from the causal world model using matching: observations in the counterfactual world that match those in a factual world are removed and replaced with the factual world for abduction. The reasoning agent (CTG-Reason) is evaluated on the generated real-world counterfactual queries.

## 2 BACKGROUND

**Structural Causal Models**   Recovering causal relationships in the real world is a long-standing problem of science (Pearl, 2009). Structural Causal Models (SCMs) (Pearl, 2009; 2014a; Xia et al., 2021) are graphical models that allow representing causal knowledge and performing causal inference. SCMs can be represented as Directed Acyclic Graphs (DAGs) where nodes are causal variables and edges are causal relationships. Causal variables can be distinguished between *endogenous* (observed) and *exogenous* (unobserved) variables. Exogenous variables $\mathbf{U}$ do not have parents and are represented by a probability distribution. Endogenous variables $\mathbf{V}$ are defined by functions that links them to their causal parents, i.e. $V_i \leftarrow f_{V_i}(\mathbf{pa}(V_i))$. SCMs are Markovian processes, which implies that if the full set of exogenous variables is

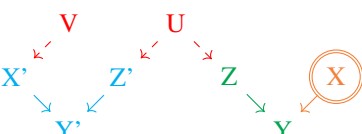

Figure 2: Counterfactual twin graph for three endogenous variables $\{X, Y, Z\}$ and two exogenous variables $\{U, V\}$, under an intervention $do(X)$. Factual and counterfactual worlds are identical except for the effect of the intervention. Exogenous variables are shared by both worlds. Intervening on $X$ removes its in-edges. $P(Y \mid do(X), X', Y')$ is obtained by estimating the value of $U$ from the factual observations, then deducing $Z$ from $U$ (left unchanged) and $Y$ from $Z$ and $X$.

known, then the values of every endogenous variable can be deterministically computed by iteratively inferring children values from their parents. While very powerful tools to estimate causal effects, SCMs typically cannot be fully retrieved as causal information is challenging to obtain.

**Counterfactual Inference** Even without an SCM, some causal questions can be answered. Causal inference queries can typically be divided into three categories: *observations*, *interventions* and *counterfactuals*, forming *Pearl's Causal Hierarchy* (Bareinboim et al., 2022). Each category is harder to answer than the previous one, as it typically requires a better understanding of the causal relationships underlying the system of interest. While multiple Markov-equivalent causal graphs can account for the same set of observations, answering interventions and counterfactuals requires one to know the true local causal structure (Bareinboim et al., 2022; Richens & Everitt, 2024). Moreover, the result of a counterfactual is generally not accessible in real-world scenarios. This is the *fundamental problem of causal inference*: only the factual world is observed (Holland, 1986).

Observational queries are represented as conditional probabilities $P(Y \mid X)$. For simplicity, we represent a single observation $X$ but this description also applies to a set of observations $\mathbf{X}$ (e.g. $P(Y|\mathbf{X})$). Interventions are represented as $P(Y \mid do(X))$ (resp. $P(Y|do(\mathbf{X}))$) using the *do-operator* (Pearl, 2009; 2012). An intervention alters the causal structure by forcing the value of the variable $X$, regardless of its prior probability, effectively cutting the parents of $X$. An intuitive example is a randomized control trial where the probability of obtaining treatment is randomized to ensure that it cannot correlate with other factors. Counterfactuals correspond to "what if?" questions, asking how would a world evolve under an intervention, given the outcome in the factual world. They can be represented with the following equation:

$$P(Y \mid do(X), X', Y') = \sum_{U \in \mathbf{U}} P(Y \mid do(X), U) P(U \mid X', Y') \tag{1}$$

$X'$ and $Y'$ represent the factual observations. $X$ and $Y$ correspond to the variables in the counterfactual world. $\mathbf{U}$ is the set of exogenous variables shared between the two worlds. Figure 2 shows a graphical expression with twin graphs corresponding to the factual and counterfactual worlds.

## 3 GRAPH RETRIEVAL-AUGMENTED CAUSAL EXTRACTION

This work introduces a method for causal extraction and reasoning based on LLM agents. This section focuses on the *Causal Cartographer Extraction Agent* (CTG-Extract). We use it to recover existing causal knowledge about the world from news sources. The resulting causal network, insights we gather from it, and its application to counterfactuals are discussed in Sections 4 and 6.

### 3.1 CTG-EXTRACT

We extract causal variables and causal relationships from news sources. Figure 1 (left) illustrates the causal extraction process. While causal structure discovery methods attempt to infer the causal structure from data, causal extraction tasks recover *stated* causal relationships in text data (Gendron et al., 2023). Large language models have shown limited performance in discovering causal relationships beyond domain knowledge (Joshi et al., 2024; Jiralerspong et al., 2024; Zecevic et al., 2023; Jin et al., 2024). Acknowledging this limitation, we rely on human sources and only use the LLM agent to perform causal extraction. Causal information can either be *explicitly* or *implicitly* stated in the

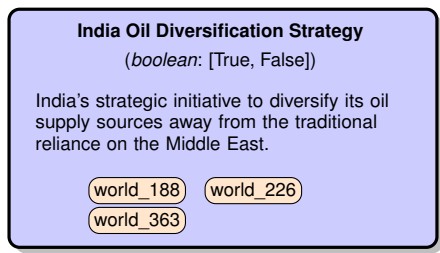

Figure 3: Example of a node extracted with CTG-Extract. Its name, description, type, possible values, and the worlds it appears in are shown.

text, within a single sentence or across multiple sentences of paragraphs. Previous methods based on pattern-matching or statistical modeling do not capture the latter well (Yang et al., 2022). Attention in LLMs allows them to aggregate information across an entire document, mitigating this problem.

**Agent Description**    We construct an LLM agent following the ReAct framework (Yao et al., 2023) and using the Smolagents library [2]. The LLM reasons with chain-of-thought (Wei et al., 2022) and provides its answer in code using a Python interpreter. A syntax check verifies if the output has the correct format and is not missing elements. If errors are identified, they are returned to the LLM. The agent can also call a retrieval-augmented generation tool called NxGraphRAG that we introduce below. The agent then adapts its next step based on this feedback. It is tasked to execute the following plan: (1) define the causal variables existing in the input text and the *confounders*, i.e. causal variables that are not observed or mentioned in the text but have a direct effect on the observed variables; (2) match the new variables with ones existing in the causal graph if possible (with NxGraphRAG); and (3) define the causal relationships between the variables based on the text, without recreating relationships already existing, and add the new causal variables and relationships to the graph.

Each variable contains the following attributes: `name` and `description`, `type` and `values` describe the domain of the variable, `current_value` is the current instantiation of the variable, `contextual_information` provides additional context to the variable instantiation and `supporting_text_snippets` is the text extract justifying the response. Each relationship contains `cause` and `effect` variables, and similar `description` and `contextual_information` attributes. Prompts and variable implementations are provided in Appendix L. An example is also provided in Figure 3.

**Grounding in Worlds**    To later perform counterfactual matching (in Section 5), we save multiple *worlds* per node. Each document describes a world, i.e. an instantiation of the observed subgraph of the world graph. For instance, two nodes $A$ and $B$ can be described in several documents. The concept they represent (described by attributes `name`, `description`, `type` and `values`) is invariant to world changes. However, their instantiations (described by attributes `current_value`, `contextual_information`, and `supporting_text_snippets`) can differ from one world to the next. They are saved as coming from different worlds, as illustrated by Figure 3. Note that confounders are by definition unobserved, so they do not have instantiations.

**NxGraphRAG**    In addition to the Python interpreter, we introduce a novel graph retrieval-augmented generation method that we equip the agent with. The system, NxGraphRAG, generates an embedding of the input document and of each node of the causal graph using an auxiliary LLM and keeps them in a vector database. Before providing the document to the agent, NxGraphRAG is called and returns the top-$K$ nodes with the highest cosine similarity to the input document. It also traverses the graph through $P$ steps from each retrieved node and adds each neighbor node and edge to ensure that no relevant context is omitted. The embedding-based retrieval focuses on **semantically relevant** nodes while the traversal takes advantage of **structural knowledge**. After this initial call, the NxGaphRAG pipeline can be called again by the agent when creating new variables to verify that they do not already exist in the graph. Additional details are provided in Appendix E.

# 4    THE CAUSALWORLD NETWORK

We use CTG-Extract to construct a causal world model that we call **CausalWorld**. It contains 975 nodes and 1337 edges. In this section, we detail the data used and the findings obtained from the network. Figure 4 shows a visualization of CausalWorld.

## 4.1    DATA

We constructed a dataset of 500 news sources describing different media events by extracting them from EventRegistry Leban et al. (2014), a platform sourcing, processing, and clustering media news based on events reported over time. Each media event is described by a title, a summary and information regarding the piece of news considered the most representative of the event, in json format. We provide the title and summary to CTG-Extract as text. We focus on media events related to economic news to avoid sensitive topics. We also focus on a restricted period of time, selecting news from 2020, to limit the effect of temporal dependencies as they are out of the scope of this study. A wide variety of economic topics are discussed in the various data samples, as illustrated in Figure 4. We use them to build a large-scale repository of causal knowledge on diverse topics.

---

[2]https://huggingface.co/docs/smolagents/

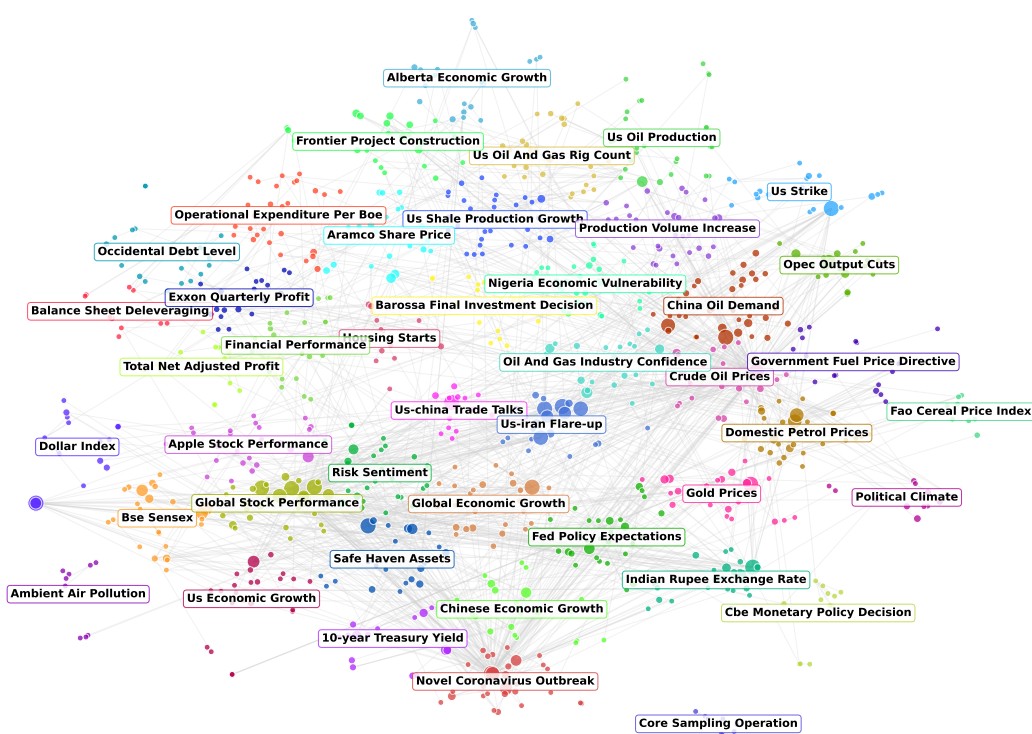

Figure 4: CausalWorld graph structure. Colors show the cluster in which the node belongs. Node size is based on the number of worlds a node appears in: the more a node appears, the bigger it is on the visualization. Labels correspond to the name of the most representative node of the cluster.

## 4.2 THE CAUSALWORLD NETWORK COMBINES MANY OVERLAPPING WORLDS

The resulting structure of CausalWorld is sparse, with a graph density of $\sim 0.001$. This is in accordance with the Sparse Mechanisms Shift (SMS) hypothesis, which states that a causal mechanism only sparsely affects other mechanisms (Schölkopf et al., 2021). Figure 11 in Appendix F further shows the distribution of strongly and weakly connected components, highlighting that the majority of the network is connected. Despite this structure, few feedback loops are observed. The network encompasses diverse knowledge and can be divided into 44 topic clusters like `Gold Prices` or `US Oil Production`, as shown in Figure 4. Furthermore, 109 structural communities can be found, communicating with other communities with a small set of bridge nodes, e.g.: `Crude Oil Prices`, `Novel Coronavirus Outbreak`, `Global Economic Growth` and `US-China Trade Talks`. These nodes are key elements of the graph that enable the propagation of information from one community to the next. More details are provided in Appendix F. The main difference between CausalWorld and other repositories of causal relationships is the presence of **worlds**, as described in Section 3. Each world instantiates a variable by providing it with a value and context. It is the key to match worlds and perform counterfactuals (discussed in Section 5). However, worlds must not be isolated and communicate via nodes shared by multiple worlds. 37% of the nodes in CausalWorld share two worlds or more, allowing information to be propagated across worlds. Since the majority of the nodes are linked by bridge nodes, most nodes can be used to compute counterfactuals.

## 4.3 CAUSAL INSIGHTS FROM STRUCTURE: "BIOFUEL DEMAND IMPACTS FOOD PRICES"

Causal paths established by the extraction process can be extracted from the network structure. We provide an example of a causal chain in Figure 5. We can observe causal relationships between nodes extracted from different documents, e.g. `Palm oil Prices` belongs to world 119 but `Food Prices` connects it to worlds 58, 68, and 70. It illustrates a key property of CausalWorld: *it allows inference to be performed between long-range dependencies across multiple sources while*

*maintaining information sparsity*. CausalWorld also allows for cycles in the graph and can, thus, represent feedback loops, as illustrated in Figure 6. More examples are provided in Appendix G.

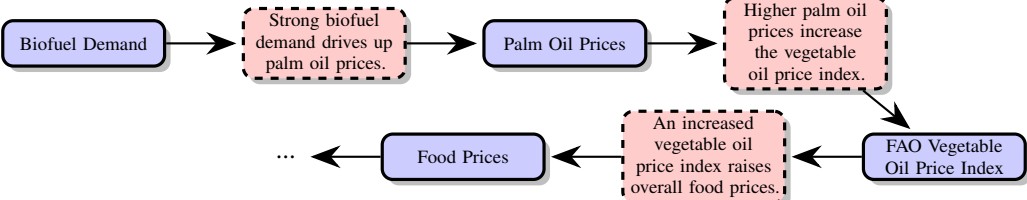

Figure 5: Illustration of a direct causal path in the CausalWorld graph. Nodes are blue boxes. Arrows represent causal dependencies. The description of the dependency is shown in dashed red boxes. Note that all nodes except for the root can have additional causal parents not shown in the chain and that the strength and function related to the causal relationships are not shown.

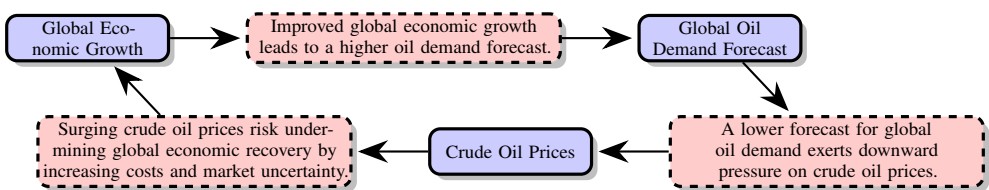

Figure 6: Illustration of a cycle in the CausalWorld graph. The legend is the same as in Figure 5.

We perform additional experiments in Appendix H to verify that CTG-Extract accurately retrieves the causal structure from the document and respects the causal relationships as stated in the data. On a synthetic benchmark, we find that CTG-Extract respects the original causal structure and, while it can miss a small fraction of causal relationships, it does not hallucinate variables or relationships.

## 5 COUNTERFACTUAL INFERENCE IN THE REAL WORLD

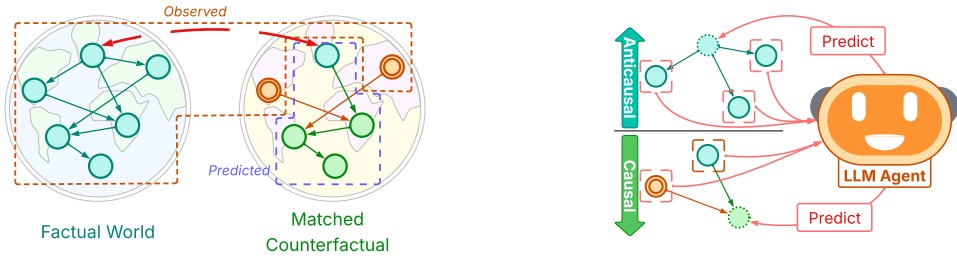

(a) Overview of counterfactual matching.    (b) Details of the step-by-step inference.

Figure 7: Overview of *counterfactual matching* and CTG-Reason. (a) A query is sampled from the CausalWorld graph using counterfactual matching: values in the counterfactual world that match those in a factual world are abducted and replaced with information from the factual world. (b) The agent performs step-by-step reasoning by predicting the value of a child variable given its parents (causal direction), resp. predicting the value of a parent given its children (anticausal).

We now investigate how to perform counterfactual reasoning in the real world. We introduce *counterfactual-matching* to obtain ground-truth counterfactual data and the *Causal Cartographer Reasoning Agent* (CTG-Reason) to perform counterfactual inference.

### 5.1 COUNTERFACTUAL MATCHING

We aim to infer the value of a target causal variable given observations or counterfactual evidence. To build such queries from CausalWorld, we use **counterfactual matching**. We define this concept and

show how it allows building real world counterfactuals. First, we must establish a **causal blanket**, i.e. a set of variables that, if known, fully determines the target variable. The causal blanket differs from a Markov blanket (Pearl, 2014b) because it only includes direct paths to the target variable and does not necessarily require the nodes in the blanket to be the target's parents. For example, in the chain $A \to B \to C \to D$, $A$ forms a causal blanket for $C$. We assume that we can construct causal blankets from the CausalWorld graph. We formally define this concept in Definition 1.

**Definition 1** (Causal Blanket). *Let $\mathcal{G}$ be a directed graphical model over a set of random variables $\mathcal{V}$, and let $T \in \mathcal{V}$ be a target variable. $\mathbf{anc}_{\mathcal{G}}(T)$ is the set of ancestors of $T$ with respect to $\mathcal{G}$. A set of variables $\mathcal{B} \subseteq \mathcal{V} \setminus \{T\}$ is called a* causal blanket *for $T$ in $\mathcal{G}$ iff, $\mathcal{B} \subseteq \mathbf{anc}_{\mathcal{G}}(T)$ and conditioned on $\mathcal{B}$, the target variable $T$ is fully determined; that is, there exists a deterministic function $f$ such that*

$$T = f(\mathcal{B}).$$

*Equivalently, knowing $\mathcal{B}$ renders $T$ conditionally independent of all other variables in $\mathcal{V} \setminus (\mathcal{B} \cup \{T\})$. Unlike the Markov blanket, a causal blanket requires that $\mathcal{B}$ contains only variables with direct causal paths to $T$, but these need not be limited to the parents of $T$. It follows that a variable can have multiple causal blankets. We analogously define causal blankets for stochastic processes in Appendix D.*

It is not generally possible to obtain the true value of a counterfactual in the real world. Therefore, we focus on a subset of counterfactual queries accessible via **K-Matching**: finding two worlds within our observations that can act as factual and counterfactual worlds. I.e. observing the factual world and intervening on $K$ observed variables of the counterfactual world is equivalent to observing the counterfactual world. Figure 7a illustrates the idea. We define this concept more formally below:

**Definition 2** (K-Matching). *Let $\mathcal{O}_o$ and $\mathcal{O}_c$ denote two sets of observations sampled from $\mathcal{V}$. Suppose that a target variable $T$ is present in both sets and that there exists a subset of shared observations $\mathcal{O}_s = \mathcal{O}_o \cap \mathcal{O}_c$ with $\mathcal{O}_s \neq \emptyset$. Let $\mathcal{B}_c \subseteq \mathcal{O}_c$ denote a causal blanket for $T$, with $|\mathcal{B}_c| = N$. We say that $\mathcal{B}_c$ can be $K$-matched with $\mathcal{O}_o$ over $T$ if it is possible to build a new causal blanket for $T$ with $K$ interventions from $\mathcal{B}_c \setminus \mathcal{O}_s$ and $N - K$ observations from $\mathcal{O}_s$.*

**Theorem 1** (K-Matching Equivalence). *Suppose that a causal blanket $\mathcal{B}_c$ is K-matched with $\mathcal{O}_o$ over a variable $T$ and that $\mathcal{O}_o \setminus \mathcal{O}_s$ forms a causal blanket over each variable of $\mathcal{O}_s$. Then, observing $\mathcal{B}_c$ or observing $\mathcal{O}_o \setminus \mathcal{O}_s$ and intervening on $\mathcal{B}_c \setminus \mathcal{O}_s$ yields the same distribution for $T$; that is,*

$$P(T \mid do(\mathcal{B}_c \setminus \mathcal{O}_s), \mathcal{O}_o \setminus \mathcal{O}_s) = P(T \mid \mathcal{B}_c).$$

The proof is given in Appendix C. This theorem implies that, if we can find a causal blanket for the counterfactual world and determine the value of $N - K$ variables of the blanket from the factual world, then we can build a counterfactual with $K$ interventions over the remaining variables.

## 5.2 CTG-REASON

We build upon the methodology proposed by Gendron et al. (2024c) to create a graph-enhanced causal inference agent. The agent follows causal inference steps described in Section 2. It performs step-by-step causal reasoning by computing the value of a variable from its direct causal parents only, or when anticausal reasoning is needed, from its direct children. At each step, the LLM only accesses the required parent/children variables. This approach, illustrated in Figure 7b, respects causal constraints and ensures that the agent only uses causal information for its reasoning, increasing efficiency and robustness by alleviating dependencies on non-causal and spurious correlations. Similarly to CTG-Extract, the agent uses the ReAct framework (Yao et al., 2023). It executes the following plan:

1. **Abduction:** If a twin graph is provided (with factual and counterfactual worlds), the agent performs abduction in an anticausal manner and computes the value of the exogenous variables of the factual world from their children. If their children are not observed, they are inferred from their own children *recursively*. The values for the exogenous variables are transferred to the counterfactual world.

2. **Intervention:** The counterfactual world is intervened upon: the incoming edges to all intervened variables are removed and the node values are fixed by the intervention value.

3. **Prediction:** The target value is inferred from its parents. If they are not observed, they are inferred from their own parents *recursively*. Since the set of observed and intervened variables forms a causal blanket, the target value is fully explained.

## 6 REAL-WORLD COUNTERFACTUAL INFERENCE EXPERIMENTS

We now take advantage of the CausalWorld graph to perform real-world counterfactual reasoning. We build a counterfactual reasoning evaluation dataset (CausalWorld-CR) from the extracted data using counterfactual-matching and evaluate CTG-Reason on the generated causal inference tasks.

### 6.1 MATCHING WITH THE CAUSALWORLD-CR DATASET

The CausalWorld-CR dataset is divided into two query types: *observations* and *counterfactuals*. For both subsets, the inference task to solve consists of inferring the value of a target causal variable given causal ancestors. **Observation queries** are created by sampling target variables and their causal blanket from a single world. The reasoning model is evaluated by inferring the value of the target variable given the values of the blanket variables. Since the query is built from a single world, the ground-truth is provided by the source document from which the world is built. The model must answer a query of the type $P(\text{Target} \mid \mathcal{B}_o)$. We use K-matching to build **counterfactual queries**. The model must answer a query of the type $P(T \mid do(\mathcal{B}_c \setminus \mathcal{O}_s), \mathcal{O}_o \setminus \mathcal{O}_s)$. We focus on 1-matching in our experiments. By default, the strategy defined above generates highly unbalanced datasets. The nodes that have a high degree and are present in multiple worlds are over-represented. We balance the dataset to reduce these effects and generate a dataset of 400 samples (see Appendix I for details).

### 6.2 EVALUATION

We perform experiments with widely used LLMs tailored for reasoning (o3-mini (OpenAI, 2025a)) and general purpose tasks (GPT-4.1 (OpenAI, 2025b) and LLaMA-3.1-8B (Meta, 2024)). We compare our step-by-step strategy against CausalCoT (Jin et al., 2023), a chain-of-thought prompting strategy (Wei et al., 2022) for causal inference. Since the evaluation queries are built from automatically scrapped real-world events, they express a diversity of types. The target variables to be predicted can be boolean, numerical or qualitative assessments of a trend. Moreover, the LLM may not provide a response corresponding to the same type. For instance, we find that LLMs tend to favor qualitative to numerical assessments (see Appendix K for more details).

**Performance** We look at the accuracy for boolean and trend queries as they correspond to categorical variables (true/false and increasing/decreasing/stable, respectively). Figure 8a shows that both strategies yield similar results, although CTG-Reason achieves slightly better accuracy. We also note that LLaMA-3.1 was not able to complete the queries with the CausalCoT strategy. We discuss this aspect in the next paragraph. Figure 8b shows the distribution of the relative error between the ground truth and the prediction for numerical counterfactual queries (in % of the ground truth value), for o3-mini and GPT-4.1. We look at the relative error instead of the absolute difference to take into account the unit difference between queries. We also excluded outliers (∼4% of the answers were nonsensical numbers, details in Appendix K). We observe that CTG-Reason and CausalCoT strategies yield similar distributions for GPT-4.1 but that CTG-Reason achieves much lower error with o3-mini (curiously, for observation queries, CausalCoT achieves better performance, see Figure 23b of Appendix K). Finally, we take a look at the semantic similarity between predicted and ground-truth answers in Figure 8c. Since CTG-Reason does not access non-causal contextual information, its wording greatly differs from the ground-truth, although this is not indicative of an incorrect answer.

**Efficiency** We study the efficiency of our proposed model in Figure 9. While CTG-Reason decomposes the problem into multiple steps and requires several LLM calls to solve a problem, this is balanced by a significantly lower quantity of retries. It can be explained by the restriction of the context to causal components, which reduces the scope of the problem and the size of the context window, *making previously intractable problems become tractable*. As seen in the previous paragraph, it does not come at the cost of a reduced performance and can greatly help small models, as illustrated with LLaMA-3.1-8B: its context size is reduced by 72% and output size is reduced by 91%.

## 7 LIMITATIONS

The iterative nature of the causal extraction implies that the processing order of the documents has an impact on the final causal network. This is not a desired behavior as the causal relationships should

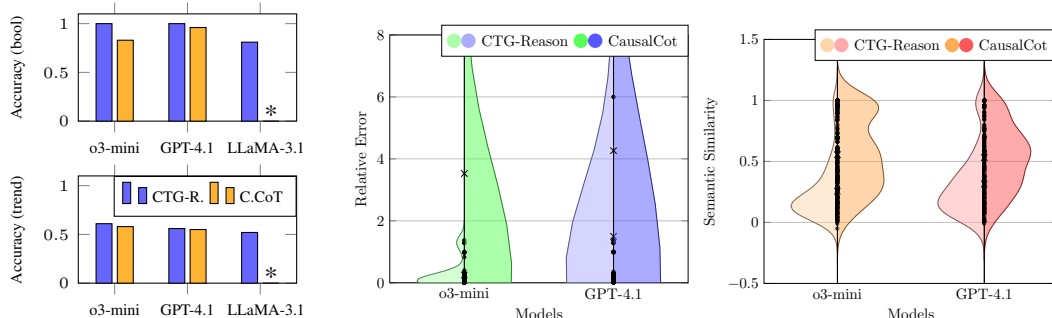

(a) Accuracy for bool/trend queries. (b) Relative error for num. queries. (c) Cosine similarity for text queries.

Figure 8: (a) Results on the boolean and trend subsets of CausalWorld-CR. Results are shown for o3-mini, GPT-4o and LLaMA-3.1-8B, using CTG-Reason (left bar/half) and CausalCoT (right bar/half). (b) and (c) Violin plots of the relative error (in %) and semantic (cosine) similarity between numerical/text ground truth and predicted answers on the counterfactual set. (*) The majority of queries with LLaMA-3.1-8B-CausalCoT returned with a timeout.

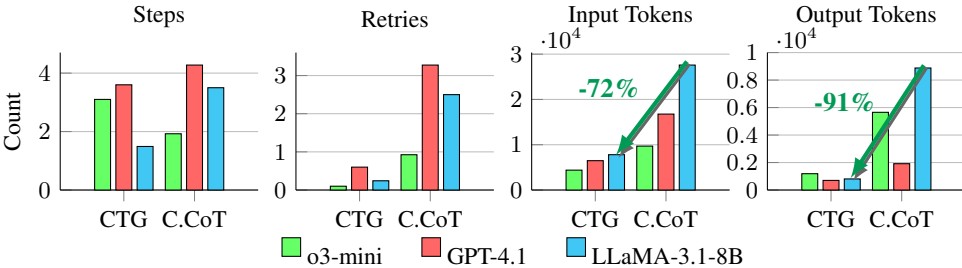

Figure 9: Statistics on the model answers with CTG-Reason and CausalCoT. (left) The average number of steps (i.e. model calls) required to solve a query. (middle left) The average number of retries after a failure to give a response (i.e. incorrect code formatting). (middle right) The average number of input tokens for the model. (right) The average number of output tokens.

be order-agnostic, potentially leading to a suboptimal configuration of the network. Moreover, since our counterfactual reasoning framework relies on causal blankets, we must assume knowledge of the full causal graph. It is not possible to guarantee that this is the case in the real world. While our method aims to reduce bias and improve robustness by retaining causal relationships instead of (potentially spurious) correlations, we rely on the accuracy and honesty of the source data and are sensitive to adversarial attacks, misinformation injections, or genuine errors. However, these issues can be mitigated by using majority voting when merging conflicting documents: a causal relationship mentioned in many documents is more trustworthy than one appearing in a single source. Other potential research directions could include combining LLMs with standard causal structure discovery methods to build the causal knowledge base. We further discuss related work and the broader impact and ethical considerations of our work in Appendices A and B.

## 8 CONCLUSION

We introduce the **Causal Cartographer**, a twofold framework composed of causal extraction and inference agents that learn causal knowledge from natural language. We use this framework to build a network of causal knowledge and prove that it allows the estimation of real-world counterfactuals. We show that our proposed step-by-step inference agent can outperform chain-of-thought baselines on counterfactual reasoning and greatly reduce the inference cost by alleviating the impact of non-causal information. We hope that our work will inspire the creation of more robust and efficient reasoning agents based on causal principles. In particular, allowing agents to learn from counterfactual information is a promising direction towards building systems of higher cognition.

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

## A    RELATED WORK

Traditional causal extraction methods rely on knowledge-based or statistical methods which can be brittle when causal relationships spread across multiple sentences or paragraphs (Yang et al., 2022). More recent approaches rely on deep learning architectures and transformers, notably CausalBERT (Khetan et al., 2021) which relies on BERT (Devlin et al., 2019), but the use of larger LLMs remains under-explored. CauseNet (Heindorf et al., 2020) is a database of causal relationships extracted from the web by mining linguistic patterns, but it does not make the distinction between variable and value that allows us to do counterfactual matching. Our NxGraphRAG method also differs from existing graph-RAG systems (e.g. GraphRAG (Edge et al., 2024)) as we do not use knowledge graphs but causal graphs. Similarly to Li et al. (2024) and Buehler (2025), we integrate the graph-RAG approach to iteratively build a graph (Peng et al., 2024). Studies have evaluated the causal reasoning abilities of LLMs, notably (Zecevic et al., 2023), (Jin et al., 2024) and Cladder (Jin et al., 2023): a comprehensive benchmark for evaluating causal reasoning in language. Unlike in our work however, Cladder is built synthetically. The term *causal blanket* has been previously introduced by Rosas et al. (2020) to characterize time-series representing dynamical systems in the domain of cognitive science. By contrast, we introduce the concept of causal blankets for directed acyclic structural causal models. The two definitions do not conflict with each other. In their work, causal blankets correspond to the mediator processes that fully capture the causal effect of a first process on the second. Our definition follows the same intuition for SCMs: a causal blanket is a set of ancestors of a variable that captures all the dependencies required to fully determine the variable.

## B    BROADER IMPACT AND ETHICAL CONSIDERATIONS

This work presents a method for building a repository of causal knowledge from real-world data in natural language, more specifically from news articles, and estimating counterfactual outcomes. We expect that it can help build better reasoning systems able to perform causal inference in natural language. The grounding in real-world data can particularly help improve the abilities of large language models on many downstream real-world tasks. Causal inference has been argued as a promising direction to reduce bias and increase fairness, trustworthiness and safety of AI systems (Pearl, 2009; Goyal & Bengio, 2020; Schölkopf et al., 2021; Bareinboim et al., 2022; Mao et al., 2022; Gendron et al., 2023; 2024a;b; Richens & Everitt, 2024) and we hope that this work can foster research in this direction and help bridge the gap with real-world applications.

However, we acknowledge that our work can also have negative impact. First, as mentioned in the limitations (Section 7), we rely on the accuracy and honesty of the source data and are not robust to adversarial attacks. Our method gives equal importance to all causal relationships, which means that a small amount of adversarial data can have a significant impact on the extraction and inference, exacerbating harmful trends and heavily affecting the downstream predictions. As a consequence, we do not recommend using our method on potentially unreliable sources. Second, our work can be used to make counterfactual predictions about real-world situations. While this work is only a first step in this direction, we expect that it will lead to economical and societal prediction engines. For instance, companies could make predictions about the economic impact of a businesses decision, or government organizations could predict the future impact of a policy on a population. Such usage can have very high impact over many areas and populations. It could provide guidance for enforcing better policies improving the well-being of populations but can also be used by malicious actors for harm. Even without the intervention of such actors, an over reliance on such prediction tools without critical analysis could similarly lead to disastrous effects in case of errors or misinterpretation of the prediction.

In its current stage, our work does not presents any such risks of misuse. However, we would like to emphasize the ethical considerations and risks links to the pursuit of this research direction and argue that future work should keep them into consideration.

## C    PROOF OF THEOREM 1

In this section, we prove the **K-Matching** theorem (Theorem 1) stated in Section 5.

*Proof.* We suppose that a causal blanket $\mathcal{B}_c$ is *K-matched* with $\mathcal{O}_o$ over a variable $T$ and that $\mathcal{O}_o \setminus \mathcal{O}_s$ forms a causal blanket over each variable of $\mathcal{O}_s$. We aim to show that $P(T \mid do(\mathcal{B}_c \setminus \mathcal{O}_s), \mathcal{O}_o \setminus \mathcal{O}_s) = P(T \mid \mathcal{B}_c)$, i.e. that a counterfactual query following this set of assumptions can be rewritten as an observational query.

We first rewrite the counterfactual query as a probability distribution over the set of exogenous variables $\mathbf{U}$, following Equation 1 (Pearl, 2009):

$$P(T \mid do(\mathcal{B}_c \setminus \mathcal{O}_s), \mathcal{O}_o \setminus \mathcal{O}_s) = \sum_{U \in \mathbf{U}} P(T \mid do(\mathcal{B}_c \setminus \mathcal{O}_s), U) P(U \mid \mathcal{O}_o \setminus \mathcal{O}_s)$$

$\mathcal{B}_c = (\mathcal{B}_c \setminus \mathcal{O}_s) \cup \mathcal{O}_s$ forms a causal blanket over $T$. Therefore, $T \perp\!\!\!\perp \mathbf{U} \mid \mathcal{B}_c$. The only paths from $\mathbf{U}$ to $T$ must go through the variables instantiating $\mathcal{O}_s$. We call them $\mathcal{S}$. We infer that $T \perp\!\!\!\perp \mathbf{U} \setminus \mathcal{S} \mid \mathcal{B}_c \setminus \mathcal{S}$. We can then rewrite the equation as follows:

$$P(T \mid do(\mathcal{B}_c \setminus \mathcal{O}_s), \mathcal{O}_o \setminus \mathcal{O}_s) = \sum_{\mathcal{S} \in \mathbf{U}} P(T \mid do(\mathcal{B}_c \setminus \mathcal{O}_s), \mathcal{S}) P(\mathcal{S} \mid \mathcal{O}_o \setminus \mathcal{O}_s)$$

We now show that $\mathcal{O}_s$ is the only possible instantiation of $\mathcal{S}$. It is assumed that $\mathcal{O}_o \setminus \mathcal{O}_s$ forms a causal blanket over $\mathcal{S}$. Therefore, $\mathcal{O}_s$ can be deterministically computed from $\mathcal{O}_o$ and the probability $P(\mathcal{S} \mid \mathcal{O}_o \setminus \mathcal{O}_s)$ will return zero probability except for the values $\mathcal{O}_s$. The equation can then be rewritten as follows:

$$P(T \mid do(\mathcal{B}_c \setminus \mathcal{O}_s), \mathcal{O}_o \setminus \mathcal{O}_s) = P(T \mid do(\mathcal{B}_c \setminus \mathcal{O}_s), \mathcal{O}_s)$$

Rule 2 of *do-calculus* (Pearl, 2009; 2012) states that an intervention can be reduced to an observation if no backdoor path connects the intervened variable to the variable of interest. It can be written as follows:

$$P(y|do(x), do(z), w) = P(y|do(x), z, w) \text{ if } (Y \perp\!\!\!\perp Z|X, W)_{\mathcal{G}_{\overline{X}\underline{Z}}} \tag{2}$$

Since $do(\mathcal{B}_c \setminus \mathcal{O}_s), \mathcal{O}_s$ forms a causal blanket over $T$, rule 2 applies and the equation can be further simplified into the desired quantity:

$$\begin{aligned} P(T \mid do(\mathcal{B}_c \setminus \mathcal{O}_s), \mathcal{O}_o \setminus \mathcal{O}_s) &= P(T \mid do(\mathcal{B}_c \setminus \mathcal{O}_s), \mathcal{O}_s) \\ &= P(T \mid (\mathcal{B}_c \setminus \mathcal{O}_s), \mathcal{O}_s) \\ &= P(T \mid \mathcal{B}_c) \end{aligned}$$

$\square$

As an illustrative example, let us imagine a graph $X \to Y \leftarrow V$ and the counterfactual $P(Y = 1|do(X = 1), X = 0, Y = 0)$. If we have two sets of observations $\mathcal{O}_o = \{X_o = 0, Y_o = 0, V_o = 0\}$ and $\mathcal{O}_c = \{X_c = 1, Y_c = 1, V_c = 0\}$, we can form $\mathcal{B}_c = \{X_c = 1, V_c = 0\}$ as these observations determine the value of $Y$ and $\mathcal{O}_s = \{V_o = V_c = 0\}$. We retrieve the above counterfactual $P(Y|do(\mathcal{B}_c \setminus \mathcal{O}_s), \mathcal{O}_o \setminus \mathcal{O}_s) = P(Y = 1|do(X = 1), X = 0, Y = 0)$.

## D    CAUSAL BLANKETS FOR STOCHASTIC PROCESSES

In this section, we define **stochastic causal blankets** that extend causal blankets to stochastic processes and prove that the **K-Matching** theorem (Theorem 1) applies to stochastic processes as long as the shared observations $\mathcal{O}_s$ can be determined from the remaining observations $\mathcal{O}_o \setminus \mathcal{O}_s$ in the factual world.

**Definition 3** (Stochastic Causal Blanket). *Let $\mathcal{G}$ be a directed graphical model over a set of random variables $\mathcal{V}$, and let $T \in \mathcal{V}$ be a target variable. $\mathbf{anc}_\mathcal{G}(T)$ is the set of ancestors of $T$ with respect to $\mathcal{G}$. A set of variables $\mathcal{B} \subseteq \mathcal{V} \setminus \{T\}$ is called a* stochastic causal blanket *for $T$ in $\mathcal{G}$ iff, $\mathcal{B} \subseteq \mathbf{anc}_\mathcal{G}(T)$ and conditioned on $\mathcal{B}$, the target variable $T$ is determined by a function $f$ and a set of exogenous (unobservable) random variables $\mathbf{U}$ such that*

$$T = f(\mathcal{B}, \mathbf{U}), \mathcal{B} \perp\!\!\!\perp \mathbf{U}$$

*and that*

$$\neg \exists V \in \mathcal{V} \text{ s.t. } V \perp\!\!\!\perp \mathcal{B} \text{ and } V \not\!\perp\!\!\!\perp T.$$

*These conditions ensure that no backdoor paths exist between $T$ and $\mathcal{B}$ via $\mathbf{U}$ and that the causal blanket covers all observables.*

Unlike in the deterministic case, observing $\mathcal{B}$ is not sufficient to determine $T$, but separates the process into its known components and the ones that can be modeled as probability distributions. This definition is less powerful, but still respects the conditions for counterfactual K-Matching.

We show the proof in the following, which unfolds analogously to the deterministic case. However, it is important to note that the proof considers a stochastic process on the variable $T$ given its stochastic causal blanket, but a standard causal blanket on $\mathcal{O}_s$. We discuss the expansion of the theorem to stochastic processes both for $T$ and $\mathcal{O}_s$ after the proof.

*Proof.* We suppose that a stochastic causal blanket $\mathcal{B}_c$ is *K-matched* with $\mathcal{O}_o$ over a variable $T$ and that $\mathcal{O}_o \setminus \mathcal{O}_s$ forms a stochastic causal blanket over each variable of $\mathcal{O}_s$. We again aim to show that $P(T \mid do(\mathcal{B}_c \setminus \mathcal{O}_s), \mathcal{O}_o \setminus \mathcal{O}_s) = P(T \mid \mathcal{B}_c)$.

We first rewrite the counterfactual query as a probability distribution over the set of exogenous variables $\mathbf{U}$, following Equation 1 (Pearl, 2009):

$$P(T \mid do(\mathcal{B}_c \setminus \mathcal{O}_s), \mathcal{O}_o \setminus \mathcal{O}_s) = \sum_{U \in \mathbf{U}} P(T \mid do(\mathcal{B}_c \setminus \mathcal{O}_s), U) P(U \mid \mathcal{O}_o \setminus \mathcal{O}_s)$$

$\mathcal{B}_c = (\mathcal{B}_c \setminus \mathcal{O}_s) \cup \mathcal{O}_s$ forms a causal blanket over $T$. In the stochastic case, we must differentiate between the variables in $\mathbf{U}$ that are blocked by the blanket and the ones that are not. The former, noted $\mathbf{U}_{\text{blocked}}$, are the same as in the deterministic case. $T \perp\!\!\!\perp \mathbf{U}_{\text{blocked}} \mid \mathcal{B}_c$. The only paths from $\mathbf{U}_{\text{blocked}}$ to $T$ must go through the variables instantiating $\mathcal{O}_s$. We call them again $\mathcal{S}$. We infer that $T \perp\!\!\!\perp \mathbf{U}_{\text{blocked}} \setminus \mathcal{S} \mid \mathcal{B}_c \setminus \mathcal{S}$. The second category, noted $U_{\text{unblocked}}$, corresponds to stochastic processes that occur on $T$ and its parents that are child variables of $\mathcal{B}_c$. The two sets are mutually exclusive.

We can then rewrite the equation as follows:

$$P(T \mid do(\mathcal{B}_c \setminus \mathcal{O}_s), \mathcal{O}_o \setminus \mathcal{O}_s) = \sum_{\mathcal{S} \in \mathbf{U}} \sum_{U_{\text{unblocked}} \in \mathbf{U}} P(T \mid do(\mathcal{B}_c \setminus \mathcal{O}_s), \mathcal{S}, U_{\text{unblocked}}) P(\mathcal{S}, U_{\text{unblocked}} \mid \mathcal{O}_o \setminus \mathcal{O}_s)$$

We now show that $\mathcal{O}_s$ is the only possible instantiation of $\mathcal{S}$, under the assumption of a deterministic process. It is assumed that $\mathcal{O}_o \setminus \mathcal{O}_s$ forms a standard causal blanket over $\mathcal{S}$. Therefore, $\mathcal{O}_s$ can be deterministically computed from $\mathcal{O}_o$ and the probability $P(\mathcal{S} \mid \mathcal{O}_o \setminus \mathcal{O}_s)$ will return zero probability except for the values $\mathcal{O}_s$. Note that in this case, a **deterministic** causal blanket is needed to guarantee the equivalence between the observation of $\mathcal{O}_o \setminus \mathcal{O}_s$ and $\mathcal{O}_s$. We discuss at the end of the proof how to extend the theorem to stochastic settings over the abducted variables. The equation can then be rewritten as follows:

$$P(T \mid do(\mathcal{B}_c \setminus \mathcal{O}_s), \mathcal{O}_o \setminus \mathcal{O}_s) = \sum_{U_{\text{unblocked}} \in \mathbf{U}} P(T \mid do(\mathcal{B}_c \setminus \mathcal{O}_s), \mathcal{O}_s, U_{\text{unblocked}}) P(U_{\text{unblocked}} \mid \mathcal{O}_o \setminus \mathcal{O}_s)$$

$U_{\text{unblocked}} \perp\!\!\!\perp \mathcal{O}_o \setminus \mathcal{O}_s$ by definition. If the two were dependent, $\mathcal{B}_c$ would no longer be a causal blanket as there would exist a variable $V \in \mathcal{O}_o \setminus \mathcal{O}_s$ s.t. $V \perp\!\!\!\perp \mathcal{B}_c$ and $V \not\!\perp\!\!\!\perp T$. So, the equation can be simplified as follows:

$$P(T \mid do(\mathcal{B}_c \setminus \mathcal{O}_s), \mathcal{O}_o \setminus \mathcal{O}_s) = \sum_{U_{\text{unblocked}} \in \mathbf{U}} P(T \mid do(\mathcal{B}_c \setminus \mathcal{O}_s), \mathcal{O}_s, U_{\text{unblocked}}) P(U_{\text{unblocked}})$$

$$= P(T \mid do(\mathcal{B}_c \setminus \mathcal{O}_s), \mathcal{O}_s)$$

The rest of the proof unfolds as in the deterministic case. Rule 2 of *do-calculus* (Pearl, 2009; 2012) states that an intervention can be reduced to an observation if no backdoor path connects the intervened variable to the variable of interest (see Eq. 2). Since $do(\mathcal{B}_c \setminus \mathcal{O}_s), \mathcal{O}_s$ forms a causal blanket over $T$, rule 2 applies and the equation can be further simplified into the desired quantity:

$$P(T \mid do(\mathcal{B}_c \setminus \mathcal{O}_s), \mathcal{O}_o \setminus \mathcal{O}_s) = P(T \mid do(\mathcal{B}_c \setminus \mathcal{O}_s), \mathcal{O}_s)$$
$$= P(T \mid (\mathcal{B}_c \setminus \mathcal{O}_s), \mathcal{O}_s)$$
$$= P(T \mid \mathcal{B}_c)$$

$\square$

As mentioned, this expansion of the theorem assumes the stochasticity of the process of determining $T$ given $\mathcal{B}_c$ but still requires that we can determine the set of abducted variables $\mathcal{O}_s$ from $\mathcal{O}_o$. Extending counterfactual matching to settings where there is not enough information to recover $\mathcal{O}_s$ is a great challenge. Exact matching cannot be performed, but a bound could be set on the error between the two matched quantities $P(T \mid \mathcal{B}_c)$ and $P(T \mid do(\mathcal{B}_c \setminus \mathcal{O}_s), \mathcal{O}_o \setminus \mathcal{O}_s)$ based on the uncertainty of the abduction process. We leave the determination of this bound for future work.

# E    DETAILS ON THE NXGRAPHRAG ARCHITECTURE

This section provides additional details regarding the graph retrieval-augmented generation pipeline introduced in Section 3. We introduce Networkx Graph Retrieval-Augmented Generation (Nx-GraphRAG), jointly used with the proposed Causal Cartographer Extraction agent (CTG-Extract). An overview of the method is shown in Figure 10. We use Langchain [3] to build the RAG pipeline. It is provided to the agent as a callable tool.

NxGraphRAG generates embeddings for the world graph nodes, candidate nodes proposed by the agent, and input documents, that can be compared using cosine similarity. We use OpenAI Embeddings (Neelakantan et al., 2022) to generate the embeddings. For the nodes, all the attributes are provided as a list of key-value character chains. The embeddings of the world graph represents the keys against which the queries, i.e. the candidate nodes and input documents, are compared with. While query vectors are generated on the fly, key vectors are stored in an vector database. We use an in-memory database because the generated world graph is small enough to fit in the memory (see Section 4 for more details). NxGraphRAG returns the top-$K$ most similar nodes with the query. It also returns the neighbors of these nodes, up to a level $P$ neighborhood. We use $K = 3$ and $P = 2$ to balance exhaustiveness and efficiency.

# F    DETAILS ON THE STRUCTURE OF CAUSALWORLD

Figure 11 shows the distribution of strongly and weakly connected components in the CausalWorld graph, highlighting that the majority of the network is connected and forms a single component. Despite this structure, few feedback loops are observed.

We now investigate the topics extracted from the news sources and represented in CausalWorld. We create clusters based on semantic embeddings. We only use the node attributes invariant to the current world. Due to the large scale of the graph, we use a smaller embedding model than in the NxGraphRAG pipeline. Since semantic clustering is only used for visualizing the graph content and is not connected to a downstream task, a downgraded performance would have no effect on the rest of our pipeline. We use all-mpnet-base-v2 (Transformers, 2024) with the SentenceTransformers

---

[3] https://www.langchain.com/

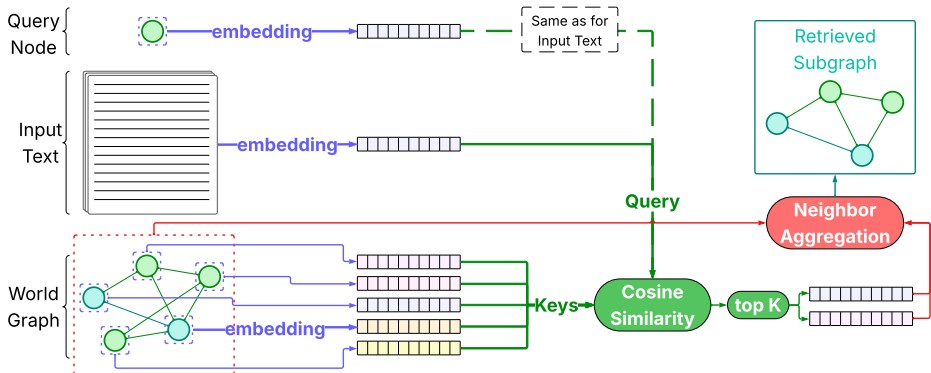

Figure 10: Overview of the NxGraphRAG pipeline. The attributes of the graph database are embedded and stored in a vector database. Depending on the use case, an input document or a candidate node are embedded as a query vector. The query vector is compared against the graph node (key) vectors using cosine similarity. The $K$ closest embeddings are selected. For each node, its $P$-level neighbors are also selected. The subgraph is returned and added to the agent context.

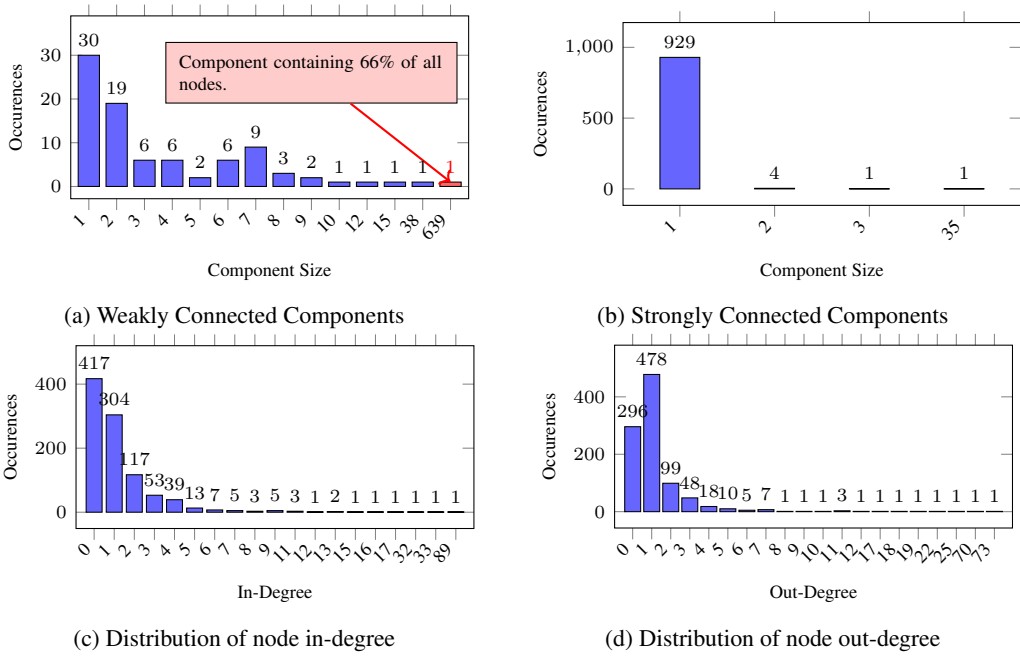

(a) Weakly Connected Components

(b) Strongly Connected Components

(c) Distribution of node in-degree

(d) Distribution of node out-degree

Figure 11: Distribution of weakly and strongly connected components and node degrees in Causal-World. The majority of nodes (66%) belong to one connected component (highlighted in red) while 3% are isolated and the others are in components of intermediate size. 5% of vertices are in non-trivial strongly connected components and are part on feedback loops. The majority of nodes have less than five neighbors, further highlighting the sparsity of the graph.

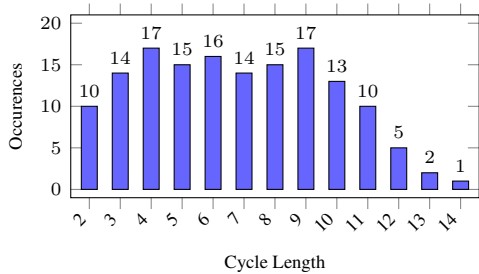

Figure 12: Distribution of cycles in CausalWorld. There are 149 cycles in the graph, with lengths comprised between two and 14.

library [4] to generate the embeddings and K-Means to attribute clusters. We use the Silhouette score (Rousseeuw, 1987) to determine the optimal number of clusters and find that the network can be best divided into 44 clusters. They can be observed in Figure 4 of the main paper.

We further use the Louvain method (Blondel et al., 2008) to detect communities within the graph based on structural information only. We discover 109 communities within the network. Eight nodes are bridge nodes separating four communities or more: `Crude Oil Prices`, `Crude Oil Prices`, `Novel Coronavirus Outbreak`, `Global Economic Growth`, `US-China Trade Talks`, `Middle East Unrest`, `US Strike`, `US-Iran Flare-up` and `US-Iran Tensions Easing`. These nodes are key elements of the graph that enable the propagation of information from one community to the next. Figure 13 illustrates the division of the network into communities.

Figure 14 shows the distribution of nodes among the worlds extracted from the input documents. We can observe that the majority of the nodes belong to a single world but that 37% of the nodes share two worlds or more, allowing information to be propagated across worlds. Since the majority of the nodes belong to the same component, most nodes can be used to compute counterfactuals.

## G ADDITIONAL CAUSAL PATHS IN CAUSALWORLD

Figure 15 shows the longest causal chain that can be extracted from CausalWorld. Additional simplified chains are shown in Figure 16.

## H EVALUATION OF THE CAUSAL EXTRACTION CORRECTNESS

This section verifies the ability of CTG-Extract to build a correct causal network on a small dataset for which ground truth is available. To this end, we build a small synthetic causal graph of twenty variables. To allow generalization, we focus on a set of variables similar to the ones discovered in CausalWorld, e.g. `oil prices`, the most observed variable. From this causal graph, we generate ten synthetic news articles describing in natural language a subset of the variables and relationships from the ground-truth graph. The news articles are generated using o3-mini-2025-01-31 (OpenAI, 2025a) to make them realistic. We manually check that the text matches the ground-truth relationships that CTG-Extract must recover from each of them.

Here an example of generated text snippet:

```
As storage capacities increase, they can mitigate short-term supply constraints and balance
    market fluctuations, ultimately exerting a decisive impact on oil pricing dynamics. [...]
```

This example describes the relationships between oil storage capacity and oil prices without directly revealing the variables and relationships that were in the ground-truth graph.

**Standard Setting** We evaluate the retrieved graph using precision, recall and F1-score. We use these measures over causal variables and relationships. For causal relationships, precision measures

---
[4]https://www.sbert.net/

1026
1027
1028
1029
1030
1031
1032
1033
1034
1035
1036
1037
1038
1039
1040
1041
1042
1043
1044
1045
1046
1047
1048
1049
1050
1051
1052
1053
1054
1055
1056
1057
1058
1059
1060
1061
1062

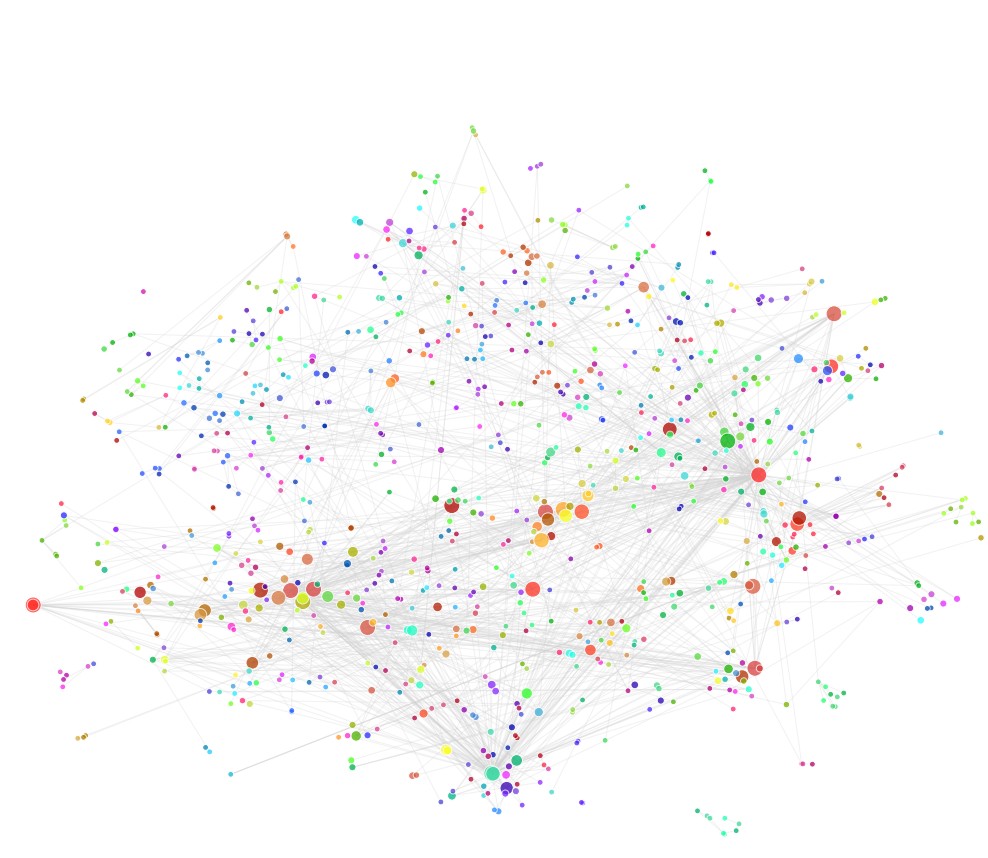

Figure 13: CausalWorld graph structure divided by Louvain communities. Colors correspond to the Louvain community in which belongs the node. Node size is based on the amount of worlds in which the node appear: more often a node appears, the bigger it is on the visualization. A total of 109 communities exists in the graph, therefore colors can correspond to several communities. This figure is for illustration purpose only.

1068
1069
1070
1071
1072
1073
1074
1075
1076
1077
1078
1079

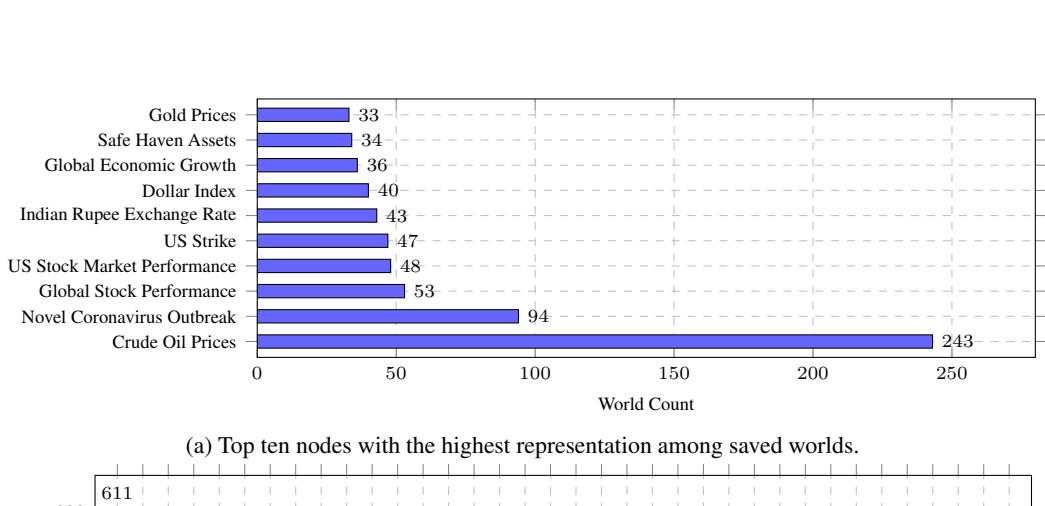

(a) Top ten nodes with the highest representation among saved worlds.

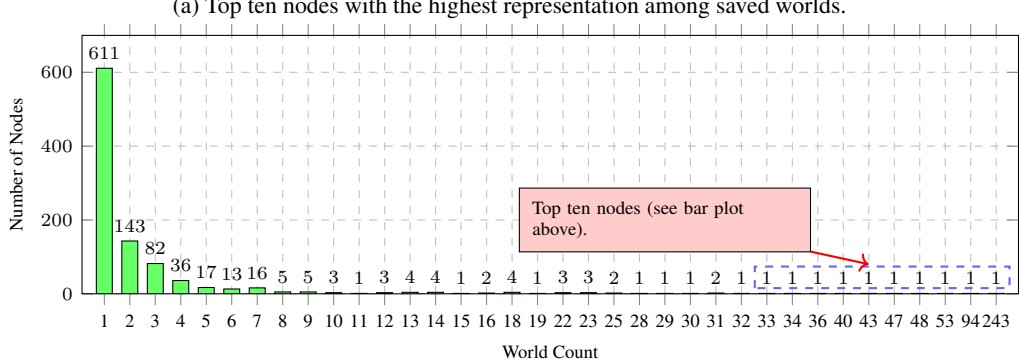

(b) Distribution of the nodes based on their number of appearances in worlds.

Figure 14: Distribution of the nodes among the worlds extracted from the source documents. The higher the world count, the most document a node appears in. The top plot shows the amount of worlds for the top ten nodes and the botom plot shows the distribution for the entire graph.

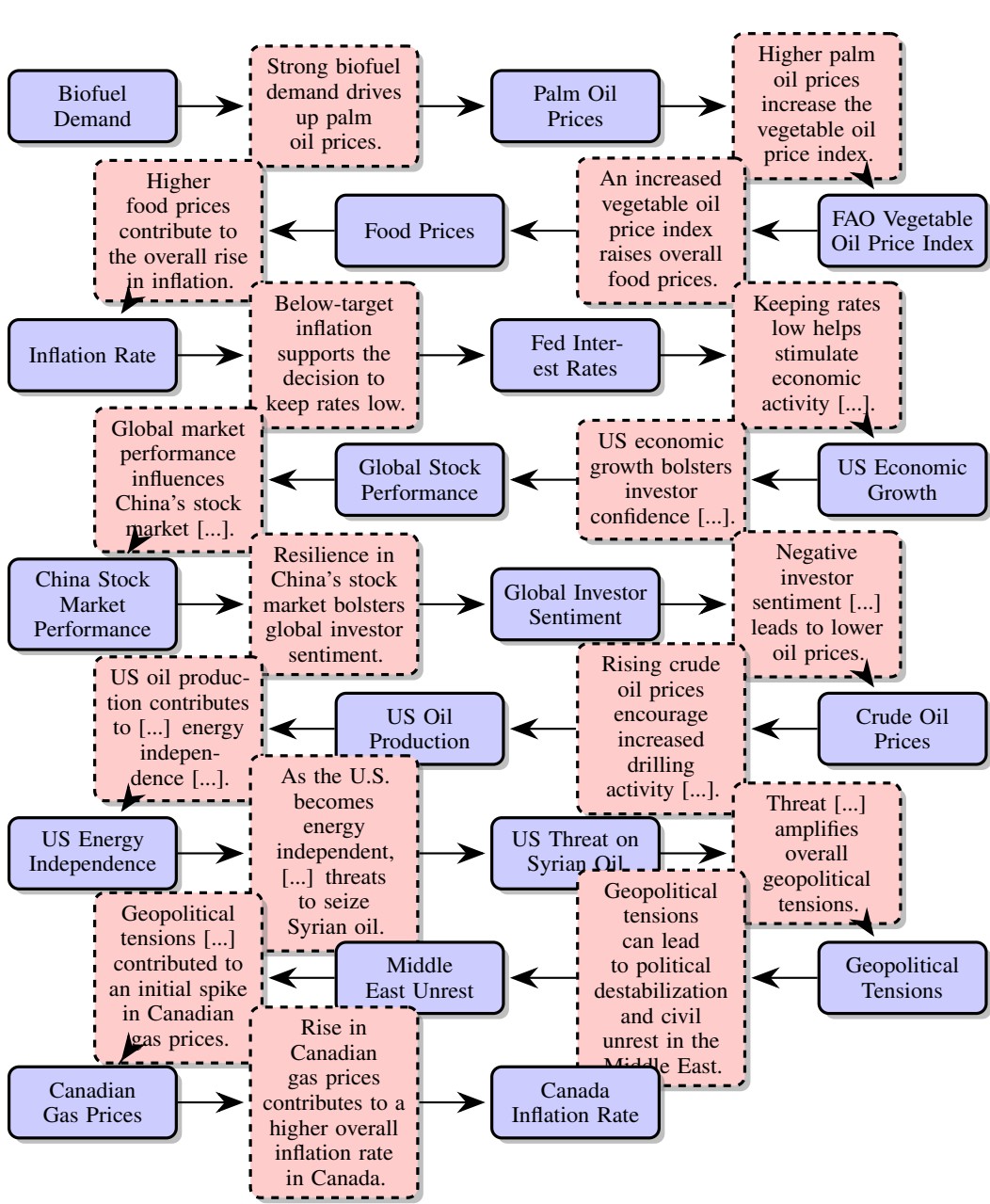

Figure 15: Illustration of the longest direct causal path in the CausalWorld graph. It contains 18 nodes, represented in blue boxes. Arrows represent causal dependencies. The description of the dependency is shown in dashed red boxes. Note that all nodes except for the root can have additional causal parents not shown in the chain and that the strength and function related to the causal relationships are not shown.

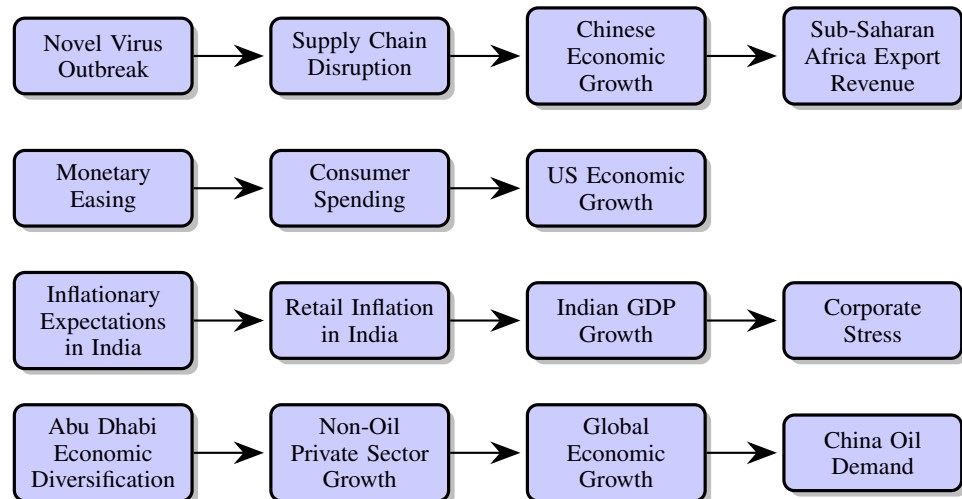

Figure 16: Additional direct causal paths in CausalWorld graph.

Table 1: Precision, recall and F1-Score of CTG-Extract on graph extraction on a synthetic graph. The metrics measure the differences between the retrieved grap hand the ground-truth synthetic graph.

|  | Precision | Recall | F1-Score |
| --- | --- | --- | --- |
| Causal Variables | 0.864 | 0.950 | 0.905 |
| Causal Relationships | 1.000 | 0.955 | 0.977 |

the proportion of true causal relationships over all retrieved relationships and recall measures the proportion of true causal relationships over all true relationships. A similar interpretation can be made for causal variables. Results are shown in Table 1.

Regarding the variables, the differences arise due to a division of one variable in the original graph into three in the extracted graph: `Investment in Exploration` is divided into `Improved Extraction Techniques`, `Oil Exploration Investments` and `Technological Investments`. Regarding the causal relationships, all retrieved relationships match the ground-truth and do not present any contradictions. A single one is missing: `OPEC Production Crude Oil Supply`. While the resulting graph differs from the ground truth, it does not present contradictions in the semantics, only some slight differences in the granularity used to describe the variables. While this experiment is performed on a small set of variables due to the difficulty in accessing real-world ground-truth causal relationships, it provides evidence that CTG-Extract accurately retrieves the original causal structure when presented with descriptive samples in plain language.

**Adversarial Setting**   To verify the robustness of our findings in adversarial scenarios, we conduct the same experiments on a causal graph with randomized edges. We aim to verify if CTG-Extract can recover the relationships from the documents if they contradict common sense. Results are shown in Table 2.

The model achieves a similar score on variable retrieval. The adversarial setting does not affect its performance. This is expected as only the edges have been randomized. However, we observe

Table 2: Precision, recall and F1-Score of CTG-Extract on graph extraction on an adversarial synthetic graph with randomized edges.

|  | Precision | Recall | F1-Score |
| --- | --- | --- | --- |
| Causal Variables | 0.905 | 0.950 | 0.927 |
| Causal Relationships | 0.981 | 0.642 | 0.777 |

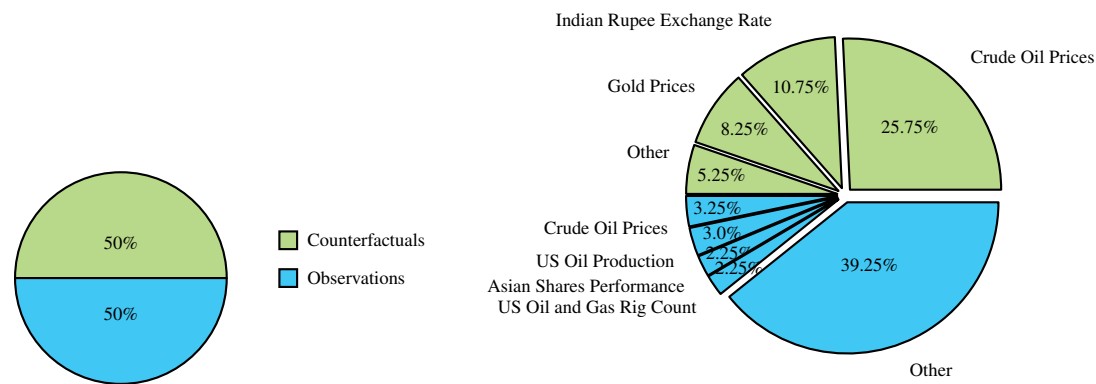

(a) Distribution of observation and counterfactual queries.

(b) Distribution of target variables in the observation and counterfactual splits.

Figure 17: Distribution of the query types and target domains in the CausalWorld-CR dataset. Counterfactual queries are more concentrated around specific domains than observations because they require to match several worlds for building the query.

a drop in performance for the retrieval of causal relationships. Precision remains very high but recall is significantly reduced: CTG-Extract has more trouble recovering some of the relationships. Nonetheless, the model does not add causal relationships that would contradict the documents. It remains accurate with respect to the original ground-truth graph. We also note that the lower recall can be due to the quality of the data: as the relationships are random, it is harder to build a coherent narrative that describes how these variables interact with each other. Here is an example of input text as illustration:

```
Recent analysis shows that subsidy regulations significantly influence alternative energy
    prices, a dynamic further complicated by refined oil product output, which in turn plays
    a pivotal role in determining both alternative energy prices and oil prices.
```

Overall, these experiments show that our proposed causal extraction method can accurately reconstruct a causal structure. The method can build a slightly different structure but does not introduce contradictory information, even in adversarial settings.

## I  DETAILS ON THE CAUSALWORLD COUNTERFACTUAL REASONING DATASET

The CausalWorld network allows matching counterfactuals and building samples for causal queries. However, as shown in Figure 11, the degree distribution in the graph is imbalanced, resulting in an over-representation of some nodes over others. We balance the dataset to mitigate this issue. In addition, in the case of long dependencies, some queries may present many causal paths between the observations and the target variable. We remove queries with too many possible causal paths (i.e. $\geq 50$) to allow the problem to remain tractable. This scenario only occurred in a small number of cases and exclusively for observational queries. Indeed, the criteria needed for counterfactual matching are harder to meet as the number of possible causal paths increases. Causal Blankets are harder to find as the number of variables increases while the number of observations remains constant. Figure 17 shows the distribution of query types and target domains in the CausalWorld-CR dataset. Figure 18 shows the number of nodes per query graph.

## J  IMPLEMENTATION DETAILS

In this section, we detail the implementation details used for our experiments.

For the causal extraction pipeline, we use OpenAI o3-mini-2025-01-31 (OpenAI, 2025a) as our base LLM. Specific details for NxGraphRAG are given in Section E. We use the Smolagents library[5]

---

[5]https://huggingface.co/docs/smolagents/

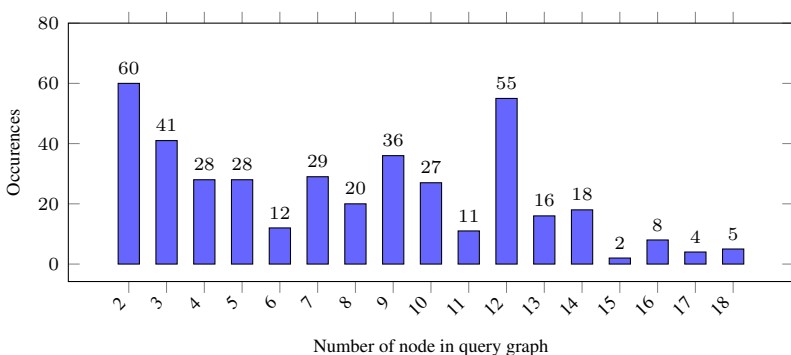

Figure 18: Distribution of the number of nodes in the query graphs of the CausalWorld-CR dataset.

Table 3: Proportion of outliers in the models evaluated on counterfactual numerical queries. The lower the better.

| Model | Proportion of outliers (%) ↓ |
|---|---|
| GPT-4.1-CTG-Reason (ours) | *7.86* |
| o3-mini-CTG-Reason (ours) | **4.45** |
| GPT-4.1-CoT | 11.17 |
| o3-mini-CoT | 11.66 |

as the interface with the LLM agents. We perform calls to o3-mini-2025-01-31 and gpt-4.1-2025-04-14 (OpenAI, 2025b) using the LiteLLM interface with default hyperparameters. The order of magnitude for the total cost of the OpenAI API calls is ∼ USD 100. We run meta-llama/Llama-3.1-8B-Instruct (Meta, 2024) locally using the Transformers interface. Local experiments are run on eight NVIDIA A100-SXM4-80GB GPUs. The data analysis performed in Sections 4 and 5 involves semantic embedding. We perform the embedding using the all-mpnet-base-v2 (Transformers, 2024) Transformer model. We use the SentenceTransformers library [6]. Data analysis experiments are performed on a single laptop with a 3.20 GHz AMD Ryzen 7 5800H CPU, 16GB RAM and a NVIDIA GeForce RTX 3070 Laptop GPU.

## K ADDITIONAL RESULTS ON CAUSALWORLD-CR

Figures 20 and 21 show the detailed results on the counterfactual and observation queries of CausalWorld-CR. The main results can be found in Figure 8. Figure 24 further shows the confusion matrices for the boolean and trend queries while Figures 22 and 23 show the distributions of the cosine similarity and relative error between the ground truth and the predictions, respectively. Figure 25 shows the BLEU scores (Papineni et al., 2002) for o3-mini and GPT-4.1 on the counterfactual and observational sets. Higher scores indicates that model responses contains n-grams similar to the ones appearing in ground truth answers. While BLEU is suited for translation, in this settings, it indicates how close the answer formulation is compared to the original. A low score does not indicate that the answer is incorrect but that the grammatical elements that are used differ. As expected, score are indeed low but observational answers obtain slightly higher scores.

Some of the numerical results for the counterfactual queries contained nonsensical outlier numbers. To do not skew the distribution, we removed them from the visualization in Figure 8b. We provide the proportion of outliers for each evaluated model in Table 3. The models using CTG-Reason are more robust and produce fewer nonsensical numbers than models with CoT, this is consistent with the results observed in Figure 8.

---

[6]https://www.sbert.net/

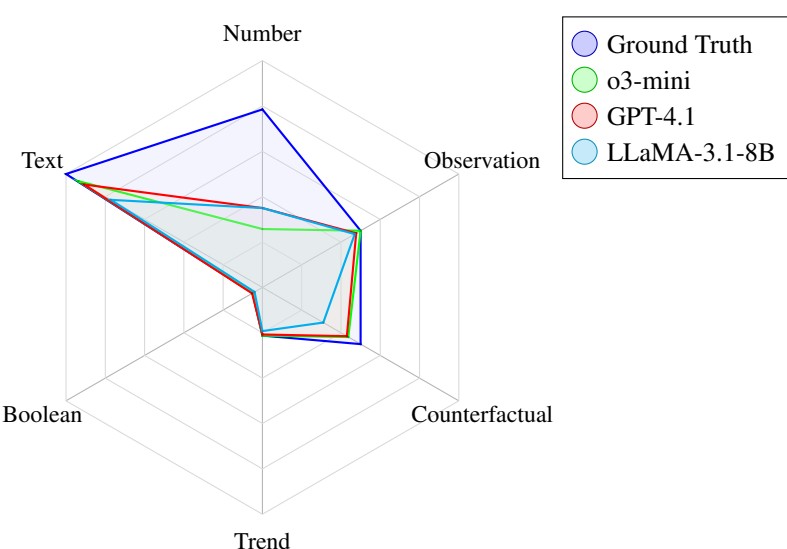

Figure 19: Distribution of query types and answer types from CTG-Reason. Types correspond to a category a sample can be evaluated in. Types are: *number* (answer is a number), *text* (answer is textual, matches all samples), *boolean* (true/false), *trend* (the qualitative assessment of a trend: increasing, decreasing, stable). *Observation* and *counterfactual* correspond to thequery types as defined i Section 5. A sample can cumulate multiple types and the model answer may not correspond to the same type as the ground truth. For instance, while the ground-truth has a numerical value, the model may provide a qualitative answer.

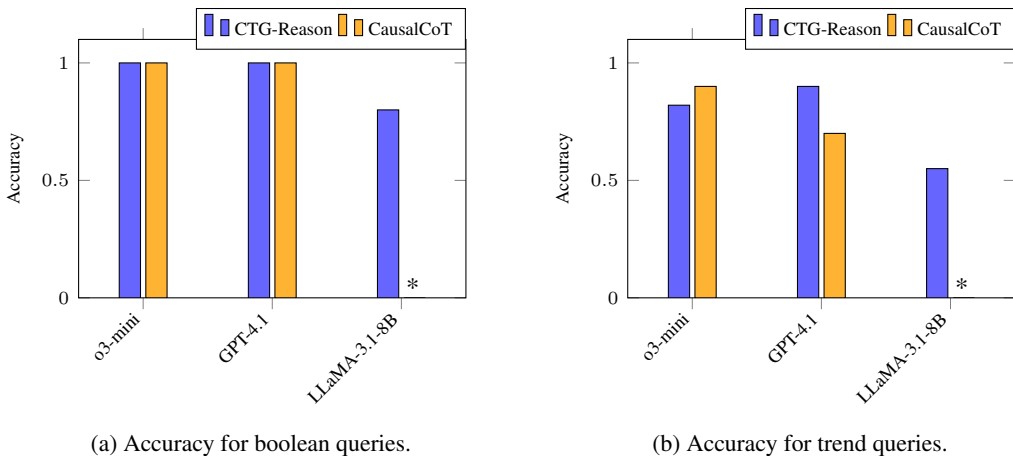

(a) Accuracy for boolean queries.     (b) Accuracy for trend queries.

Figure 20: Results on the *counterfactual* set for the boolean and trend queries of CausalWorld-CR. Results are shown for o3-mini, GPT-4o and LLaMA-3.1-8B, using CTG-Reason and CausalCoT. (*) The majority of queries with LLaMA-3.1-8B-CausalCoT returned with a timeout.

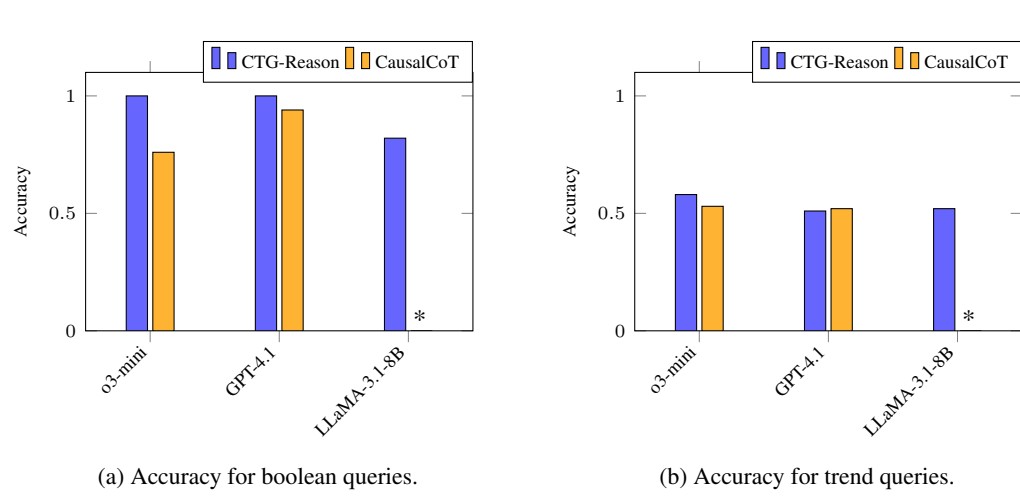

(a) Accuracy for boolean queries.

(b) Accuracy for trend queries.

Figure 21: Results on the *observational* set for the boolean and trend queries of CausalWorld-CR. Results are shown for o3-mini, GPT-4o and LLaMA-3.1-8B, using CTG-Reason and CausalCoT. (*) The majority of queries with LLaMA-3.1-8B-CausalCoT returned with a timeout.

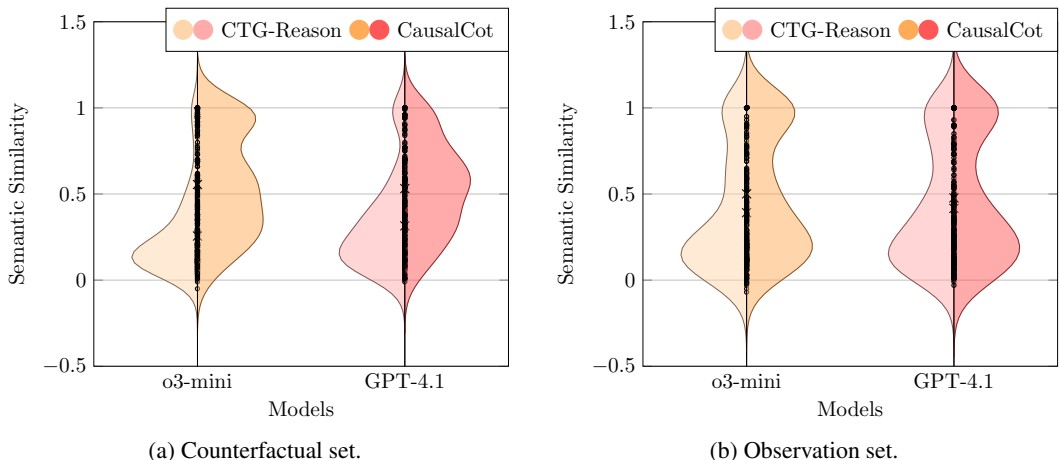

(a) Counterfactual set.

(b) Observation set.

Figure 22: Violin plots of the semantic (cosine) similarity between ground truth and model answers. Results are shown for o3-mini and GPT-4.1.

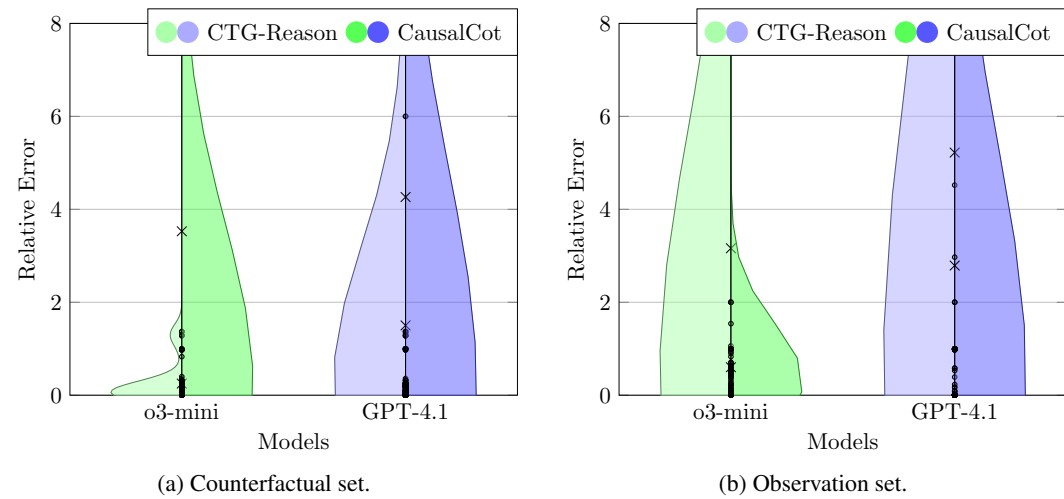

Figure 23: Violin plots of the relative error numerical between ground truth and prediction. The results are shown for o3-mini and GPT-4.1 for samples where both ground truth and predicted values are numerical. The plots show the distribution of the error in % of the ground truth value.

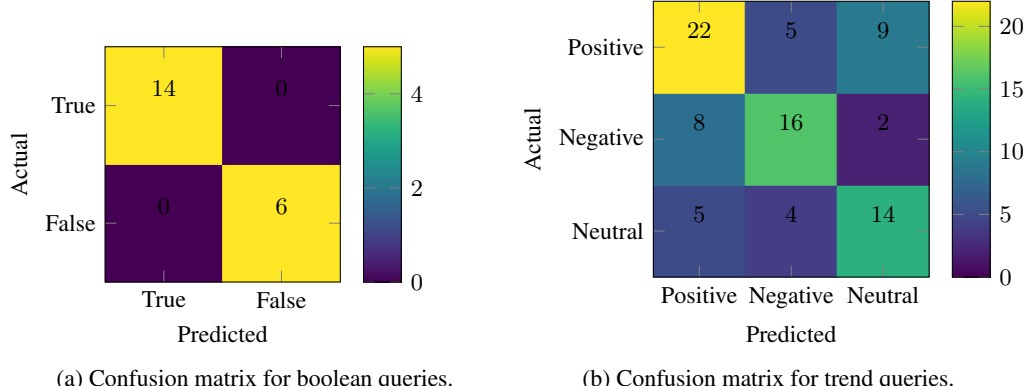

(a) Confusion matrix for boolean queries.

(b) Confusion matrix for trend queries.

Figure 24: Confusion matrices for boolean and trend queries of CausalWorld-CR for o3-mini using CTG-Reason.

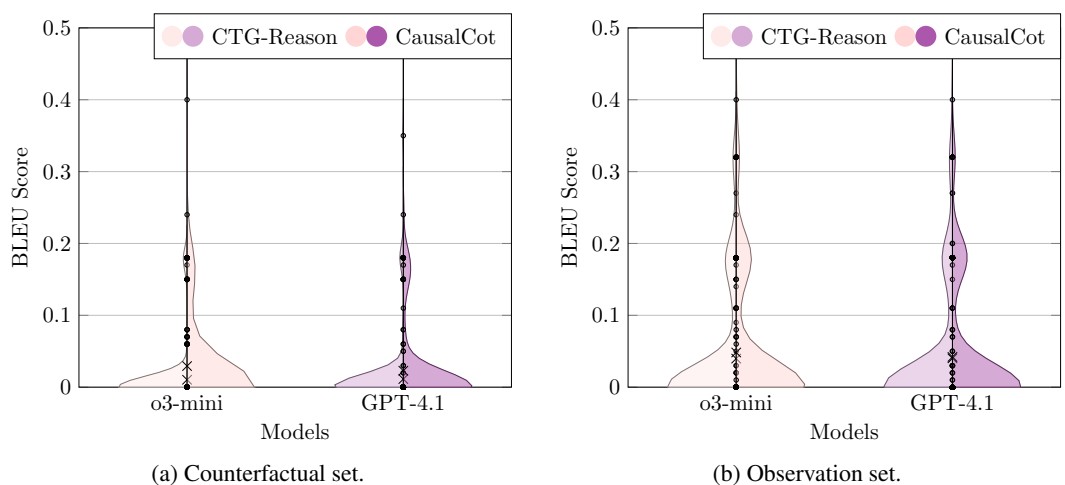

(a) Counterfactual set.

(b) Observation set.

Figure 25: Violin plots of the BLEU score for o3-mini and GPT-4.1 using CTG-Reason on counterfactual and observational query sets. A higher score indicates that the model prediction contains n-grams similar to the ones appearing in the ground truth answer.

## L    AGENT PROMPTS

This section describes the system and user prompts used for the LLM agents in the extraction and inference systems, as well as the syntax for the causal variables and relationships in the causal network.

### L.1    CAUSAL INFORMATION SYNTAX

The syntax for the causal variables and relationships is provided in the LLMs' prompts as shown in the next sections. Additionally, it is also enforced using Pydantic.

#### L.1.1    CAUSAL VARIABLES

Causal variables are defined in the causal network as a dictionary of attributes. They are described to the LLM agents as follows:

```
{
    "name": <string>, # The name of the variable
    "description": <string>, # The description of the variable
    "type": <string>, # The type of the variable (boolean, integer, float, string, etc.)
    "values": <List[str]>, # The set or range of possible values for the variable ([1, 2, 3], '
        range(0,10)', ['low', 'medium', 'high'], 'True/False', 'natural numbers', etc.)
    "causal_effect": <string>, # The inferred causal effect of the variable
    "supporting_text_snippets": <Optional[List[str]]>, # The supporting text snippets in which
        the variable is mentioned
    "current_value": <Optional[string]>, # The observed current value of the variable
    "contextual_information": <Optional[string]>, # The contextual information associated with
        the current value of the variable
}
```

See Section 3 in the main paper for more details.

#### L.1.2    CAUSAL RELATIONSHIPS

Similarly to causal variables, causal relationships are defined in the causal network as a dictionary of attributes. They are described to the LLM agents as follows:

```
{
    "cause": <string>, # The name of the cause variable
    "effect": <string>, # The name of the effect variable
    "description": <string>, # The description of the causal relationship between the variables
    "contextual_information": <Optional[string]>, # The contextual information associated with
        the causal relationship for the specific observed values of the variables
    "type": <string>, # The type of the causal relationship (direct, indirect, etc.)
    "strength": <Optional[string]>, # The strength of the causal relationship
    "confidence": <Optional[string]>, # The confidence level in the existence of the causal
        relationship
    "function": <Optional[Callable]>, # The function that describes the causal relationship, if
        available.
}
```

### L.2    CAUSAL EXTRACTION AGENT PROMPT

We show below the start of the causal extraction agent system prompt. It describes to the agent the extraction task to be performed and how to solve it. It is built upon on the ReAct framework (Yao et al., 2023) and the prompts provided by the Smolagents library [7].

```
You are an expert assistant who can solve any task using code blobs. You specialize into
    causal extraction tasks.
You will be given a text snippet and an initial causal graph. Your task will consist of
    finding the causal variables described in the text, the causal relationships that link
    them and adding them to the causal graph.
You will solve the task as best you can. To do so, you have been given access to a Python
    interpreter with the standard library and the networkx package.
You will also have access to an optional list of tools: these tools are basically Python
    functions which you can call with code.
```

---

[7]https://huggingface.co/docs/smolagents/

```
You will use your expert reading comprehension, commonsense reasoning and coding skills to
    tolve the problem.
To solve the task, you must plan forward to proceed in a series of steps, in a cycle of '
    Thought:', 'Code:', and 'Observation:' sequences.

At each step, in the 'Thought:' sequence, you should first explain your reasoning towards
    solving the task and the tools that you want to use.
Then in the 'Code:' sequence, you should write the code in simple Python. The code sequence
    must end with '<end_code>' sequence.
During each intermediate step, you can use 'print()' to save whatever important information
    you will then need.
These print outputs will then appear in the 'Observation:' field, which will be available as
    input for the next step.
In the end, you have to return a final answer using the 'final_answer' tool. The output
    provided to the 'final_answer' tool should be the networkx causal graph.

Each node should have the following dictionary of attributes:
{{variable}}
Some variables will have a value provided in the text, while others will be confounders that
    need to be estimated. Provide a current~value and contextual information whenever
    possible.

The causal relationships should be represented as directed edges between the nodes. Each edge
     should have the following dictionary of attributes:
{{causal_relationship}}

Your plan should be as follows:
1. Define the causal variables observed in the text. Use the variables provided when possible
     or create new ones when no variable matches.
2. Define the confounders that are not observed in the text or for which a value is not given
    , and that affect one or several of the causal variables defined in step 1.
3. Verify if the new variables have correspondance in the causal graph database. Use the '{{
    retrieval_tool_name}}' tool to assess if the variable is already in the database.
If it is, use it instead of creating a new one. It may have a different name in the database,
     the tool returns the top-k matching variables.
THIS IS A MANDATORY STEP. The variables provided in the input are only a subset of the
    variables in the database, you should always check if the variable already exist before
    creating new ones.
For each variable, use the one matching the most or create a new one if none matches.
4. Define the causal relationships between the variables, based on the text and common sense
    knowledge. Do not create causal relationships that already exist in the causal graph.
5. Build the full causal graph as a networkx DiGraph object.
Each step should be a separate 'Thought:', 'Code:', and 'Observation:' sequence.

The code MUST be executed in two code blocks minimum. After step 3, use <end_code> to
    indicate the end of the code block and retrieve the output of the tool call.
Then, use the observation to execute steps 4 and 5 and complete the task. An empty list of
    variables in the user input DOES NOT MEAN that there are no variables in the causal
    graph database.
Always check if the variable exists in the database before creating a new one.

[...]
```

The `variable` and `causal_relationship` placeholders are replaced with the prompts described in the previous section. Few-shots are also provided as part of the system prompt. Here is one example:

```
[...]

Here are a few examples tasks to solve.
---
Task:
"""
Increased Air Pollution Leads to Higher Respiratory Issues During Summer Months

A recent study has shown that cities experiencing high levels of air pollution are seeing a
    rise in respiratory issues among their populations, particularly during the summer months.

Experts attribute the increase to a combination of higher temperatures and pollution from
    industrial activities, which intensify during hotter months.
Dr. Laura Simpson, a pulmonologist, noted that individuals with pre-existing conditions, such
    as asthma, are at a greater risk of suffering from exacerbated symptoms.

Retrieved nodes:
<empty>

Retrieved edges:
<empty>
"""
```

```
Thought: I will first define the list of causal variables extracted from the text. These
    include:
1. Air Pollution - representing high levels of air pollution in cities.
2. Temperature - higher temperatures during the summer months.
3. Industrial Pollution - arising from intensified industrial activities during hotter months.
4. Respiratory Issues - the rise in respiratory issues among the population.
5. Pre-existing Respiratory Conditions - representing vulnerable individuals (e.g., those with
    asthma) at greater risk.

I will verify if these variables exist in our causal graph database using the tool
    graph_retriever for each variable by calling it with a query that includes the variable
    name. Then I will build the networkx DiGraph with these nodes and add the appropriate
    directed edges based on cause-effect relationships:
- Air Pollution → Respiratory Issues ("Increased air pollution leads to higher respiratory
    issues.")
- Temperature → Industrial Pollution ("Hotter temperatures intensify industrial activities'
    pollution.")
- Industrial Pollution → Respiratory Issues ("Pollution from industrial activities
    contributes to respiratory issues.")
- Pre-existing Respiratory Conditions → Respiratory Issues ("Individuals with pre-existing
    conditions are more vulnerable to respiratory issues.")

I will now call graph_retriever to check for each variable.

Code:
```py
import networkx as nx

# Define initial variables extracted from the text
causal_variables = [
{
    "name": "Air Pollution",
    "description": "The level of air pollutants present in the city environment.",
    "type": "float",
    "values": "range(0,100)",
    "supporting_text_snippets": ["cities experiencing high levels of air pollution"],
    "current_value": None,
    "contextual_information": "High in some cities as noted in the study."
},
{
    "name": "Temperature",
    "description": "Ambient temperature, particularly during the summer months.",
    "type": "float",
    "values": "range(20,50)",
    "supporting_text_snippets": ["higher temperatures", "summer months"],
    "current_value": None,
    "contextual_information": "Elevated during summer."
},

(truncated)

]

# Query the causal graph database for each variable using graph_retriever
for variable in causal_variables:
query_text = f"{variable['name']}: {variable['description']}"
result = graph_retriever(query=query_text)
print(f"Query for variable '{variable['name']}':", result, '\n')

print("End of variable verification.")
```<end_code>
Observation:
Query for variable 'Air Pollution': Retrieved nodes:
<empty>

Retrieved edges:
<empty>

Query for variable 'Temperature': Retrieved nodes:
<empty>

Retrieved edges:
<empty>

(truncated)
```

```
End of variable verification.

Thought: None of our causal variables exist in the database, so I will now build the full
    networkx DiGraph using our defined variables (as nodes) and add the causal relationships
    (as directed edges).
I will add the following edges:
- Air Pollution --> Respiratory Issues: "Increased air pollution leads to higher respiratory
    issues."
- Temperature --> Industrial Pollution: "Higher temperatures intensify industrial pollution
    from industrial activities."
- Industrial Pollution --> Respiratory Issues: "Pollution from industrial activities
    contributes to respiratory issues."
- Pre-existing Respiratory Conditions --> Respiratory Issues: "Individuals with pre-existing
    respiratory conditions are more vulnerable to respiratory issues."

I will now add these nodes and edges to the graph and return the final causal graph.

Code:
```py
# Build the full causal graph as a networkx DiGraph
G = nx.DiGraph()

# Add nodes for each variable and update their attributes separately
for variable in causal_variables:
var_name = variable["name"]
G.add_node(var_name)
G.nodes[var_name].update(variable)

# Define causal relationships as edges with descriptive attributes
causal_relationships = [
{
    "cause": "Air Pollution",
    "effect": "Respiratory Issues",
    "description": "Increased air pollution in cities leads to higher respiratory issues among
        populations.",
    "contextual_information": "Study shows a rise in respiratory issues in high pollution
        cities during summer.",
    "type": "direct",
    "strength": "high",
    "confidence": "high",
    "function": None
},
{
    "cause": "Temperature",
    "effect": "Industrial Pollution",
    "description": "Higher temperatures intensify industrial activities, leading to increased
        industrial pollution.",
    "contextual_information": "Pollution from industrial activities intensifies during hotter
        months.",
    "type": "direct",
    "strength": "moderate",
    "confidence": "high",
    "function": None
},

(truncated)

]

# Add edges from the causal_relationships list
for edge in causal_relationships:
cause = edge["cause"]
effect = edge["effect"]
G.add_edge(cause, effect)
G[cause][effect].update(edge) # /!\ This is how edge attributes should be added. They must be
    added after the edge is created.

# Output the final graph details for verification
print("Causal Graph Nodes:")
print(list(G.nodes(data=True)))
print("\nCausal Graph Edges:")
print(list(G.edges(data=True)))

final_answer(G)
```<end_code>

[...]
```

Three examples are provided to the LLM. The rest of the prompt is identical to the default system prompt of Smolagents.

## L.3 COUNTERFACTUAL INFERENCE AGENT PROMPT

We show below the start of the counterfactual inference agent system prompt, it is built similarly to the causal extraction agent.

```
You are an expert assistant who can solve any task using code blobs. You specialize into
    causal inference tasks.
You will be given causal variables and causal relationships. Your task will consist of
    computing the causal effect of a target causal variable given the values of its direct
    parents or children in the causal graph.
You will solve the task as best you can. To do so, you have been given access to a Python
    interpreter with the standard library.
You will also have access to an optional list of tools: these tools are basically Python
    functions which you can call with code.
You will use your expert reading comprehension, commonsense reasoning and coding skills to
    tolve the problem.
To solve the task, you must plan forward to proceed in a series of steps, in a cycle of '
    Thought:', 'Code:', and 'Observation:' sequences.

At each step, in the 'Thought:' sequence, you should first explain your reasoning towards
    solving the task and the tools that you want to use.
Then in the 'Code:' sequence, you should write the code in simple Python. The code sequence
    must end with '<end_code>' sequence.
During each intermediate step, you can use 'print()' to save whatever important information
    you will then need.
These print outputs will then appear in the 'Observation:' field, which will be available as
    input for the next step.
In the end, you have to return a final answer using the `final_answer` tool. The output
    provided to the `final_answer` tool should be the networkx causal graph.

The attributes of the target variable are provided as arguments with the name 'target_variable
    '.
The parent variables attributes are provided as a list of dictionaries with the name '
    parent_variables'.
The children variables attributes are provided as a list of dictionaries with the name '
    children_variables'.
The descriptions of the causal relationships between the target variable and its parents are
    provided as a list of attribute dictionaries with the name 'causal_relationships'.
Return a dictionary with the updated attributes of the target variable. The fields to update
    are 'current_value', 'contextual_information' and 'causal_effect'
Use the information from the given target variable, parent variables, children variables and
    causal relationships to compute the causal effect of the target variable.
When a function is provided in the causal relationship, execute it with the Python interpreter.

Each causal variable will be represented with the following dictionary of attributes:
{{variable}}

The causal relationships will be represented with the following dictionary of attributes:
{{causal_relationship}}
```

Five few-shots are also provided as part of the system prompt. Here is one example:

```
[...]

Here are a few examples tasks to solve.
---
Task:
"""
Compute the causal effect of the variables in the context of the oil price war

You have been provided with these additional arguments, that you can access using the keys as
    variables in your python code:
{'children_variables': [], 'target_variable': {'name': 'oil_prices', 'description': 'Global
    price per barrel of oil; recently slumped to its lowest level since 2009', 'type': 'float
    ', 'values': 'USD per barrel'}, 'causal_relationships': []}.
"""

Thought: We have a target variable "oil_prices" with description indicating that oil prices
    have slumped to the lowest level since 2009.
There are no parent or children variables provided and no causal relationships specified.
In this context, we need to update the target_variable dictionary by setting its "
    current_value", "contextual_information", and "causal_effect".
```

```
Since the variable is operating in an environment described as an "oil price war", we can
    infer that the aggressive competition among oil producers is likely driving a strong
    negative causal impact on oil_prices.

I'll now update the target variable accordingly:
- current_value: "lowest level since 2009"
- contextual_information: "In the context of an oil price war, aggressive market competition
    among oil-producing nations has driven global oil prices down significantly."
- causal_effect: "lowest level since 2009"

I'll now write the code with these updates and then use final_answer to output the updated
    target_variable.

Code:
```py
# Update target_variable with computed values
target_variable['current_value'] = "lowest level since 2009"
target_variable['contextual_information'] = ("In the context of an oil price war, aggressive
    market competition among oil-producing nations has driven global oil prices down
    significantly.")
target_variable['causal_effect'] = "lowest level since 2009"

final_answer(target_variable)
```<end_code>

[...]
```

