# OpenReview forum: "Causal Cartographer: From Mapping to Reasoning Over Counterfactual Worlds"
_ICLR.cc/2026/Conference — Submitted to ICLR 2026_

### Official Review · Reviewer_6vkm · 2025-10-23

**Soundness:** 2
**Presentation:** 3
**Contribution:** 3
**Rating:** 4
**Confidence:** 2

**Summary:**

It is difficult to both perform causal reasoning with large language models and evaluate them, due to ladder of causality wich states that interventional and counterfactual quantities can generally not be inferred from observed data unless strong assumptions. The submission proposes to extract known causal relationships from real-world articles, yielding a causal world model, then use that model to perform causal reasoning with LLMs as well as evaluating them.

**Strengths:**

- Original approach, which I had not seen before while seeming natural in the context of LLMs.

- The paper is generally clear and well-written.

- Experiments support the method outperforming past alternatives in terms of performance and efficiency.

**Weaknesses:**

- The definition of SCMs used by authors ignores noise in individual structural equations. Notably, Definition 1 assumes deterministic relationships between the causal blanket and the target variable, while in general, noise variables can be present. This makes it unclear to assess whether Definition 1 and Theorem 1 is only possible in the absence of noise, which is a generally restrictive scenario.

- "We also excluded outliers (∼4% of the answers were nonsensical numbers)." (l.413-415) : this seems a bit quick to me... It would helpful to know the fraction of outliers for each evaluated model, how they change results, and how to evaluate performance in a way that is robust to them if they dominate averages.

**Questions:**

- How do Definition 1 and Theorem 1 generalize in the presence of noise variables?

- What if you include outliers, and check the things indicated above?

---

> ### Author Response · Authors · 2025-11-20
>
> We thank Reviewer 6vkm for their feedback. Regarding your concerns and questions:
> - As mentioned in the response to Reviewer ksfa, the current version of the theorem is compatible with noise and stochastic processes. We follow the SCM definition of causal processes, which represent stochasticity as random variables. In this theoretical framework, causal processes are deterministic functions of endogenous and exogenous variables, and the latter may include random processes. We use the same representation in our work and allow stochastic mechanisms.
> - We removed these outliers to facilitate interpretation of the distribution of numerical responses in Figure 8 (b), as including the outliers would heavily skew the distribution to extreme values. In the paper, we provided the exclusion number for the model with the fewest outliers, namely o3-mini-CTG-Reason. We provide below the detailed proportion of outliers for each model in Figure 8 (b):
>
> | Model | Proportion of outliers (%) $\downarrow$ |
> |:--------|:--------------------------------|
> | GPT-4.1-CTG-Reason (ours) | *7.86* |
> | o3-mini-CTG-Reason (ours) | **4.45** |
> | GPT-4.1-CoT | 11.17 |
> | o3-mini-CoT | 11.66 |
>
>
> Overall, the models using CTG-Reason are more robust and produce fewer nonsensical numbers than models with CoT, this is consistent with the results observed in Figure 8. We will include these details in the latest version of the paper.

---

> ### Comment · Reviewer_6vkm · 2025-11-25
>
> Thanks for the rebuttal. The outliers seem solved to me. However, I am still convinced that the definition of a causal blanket should be relaxed to stochastic relationships. Indeed, albeit an SCM does technically incorporate noise variables, Definition 2 shows that causal blankets and variables under scrutiny are all supposed to be **observable**, while a noise variable actually is not. This remains the key weakness of the theory. Can you adapt the definition of a causal blanket to only observable variables, and not noise variables?

---

> > ### Author Response · Authors · 2025-11-29
> >
> > Thank you very much for your response and engagement. Definition 2 assumes that the variables to be matched must be observable and observed in both the factual and counterfactual worlds but this requirement does not apply to other variables in the causal graph. Therefore, the notion of causal blankets can be easily extended to account for stochastic processes where noise variables can influence the relationships between the causal blanket and the target variable. We have expanded our definition of causal blankets to stochastic processes in the latest revision of the paper to take this case into account. This revision also includes the analysis of outliers discussed above. We hope that these improvements have answered your remaining concern.

---

### Official Review · Reviewer_F6Ef · 2025-11-01

**Soundness:** 3
**Presentation:** 3
**Contribution:** 3
**Rating:** 4
**Confidence:** 4

**Summary:**

The authors present Causal Cartographer which extracts causal relationships from data and then uses them as constraints to perform step-by-step causal inference. The performance is evaluated on real-world counterfactuals obtained from news sources.

**Strengths:**

1. Understanding LLM performance on counterfactual reasoning tasks is crucial is furthering reasearch on LLMs ability to do causal tasks
2. Using real-world data instead of synthetic is encouraging
3. The proposed method is more interpretable which is good for future research

**Weaknesses:**

1. It is not clear how this method can scale to production LLM systems.
2. Causal graph building would require a very careful control so as to not introduce bias

**Questions:**

1. Does this framework have the risk of running "stale". In a constantly evolving world, what if the causal relationships from the first stage change? How would one go about keeping them up to date? Would this update process eat into the inference cost savings?
2. How does the system defend itself against adversarial attacks where noisy/false claims are injected into the causal knowledge repository?
3. Sorry if I missed this, but how is it ensured that the extracted relationships are causal and not noise?

---

> ### Author Response · Authors · 2025-11-20
>
> We thank Reviewer F6Ef for their feedback. Regarding your concerns:
> - Our method takes advantage of state-of-the-art LLMs while using a lower computational budget than standard chain-of-thought. We also apply our work on real-world data. While we do not claim that our work can be directly used in production systems, we provide evidence that it performs as well as CoT models for a much lower budget on challenging natural language causal inference tasks. We believe it can significantly help performing causal inference with LLMs in the wild.
> - Could further elicit what you mean by “careful control”? Causal graphs provide greater guarantees of unbiasedness compared to standard correlation-based models because causal graphs flatten the information into causal relationships, removing the spurious effects of over- and under-sampling some domains: a causal relationship is in the graph once whether it appears once in the data or multiple times, allowing minority representations to be effectively represented. Causal relationships are also more robust to bias as they disambiguate correlations from causation [1,2].
> However, one assumption that one needs to make is the truthfulness of the data. In our work, we use high-quality data to ensure that the extracted causal relationships can be trusted. However, even in the presence of noisy or incorrect information, it is not a bottleneck in the application of the method as additional safeguards can be applied at the extraction step using standard causal structure discovery methods to ensure the validity of the extracted causal relationships. We discuss this idea further in Section 7.
>
> Regarding your questions:
> - This is an interesting question. We did not consider staleness in our work, but this issue can be easily addressed. Indeed, a causal graph is an interpretable representation of knowledge, with one subpart that can be edited without affecting other parts of the graph (due to the sparsity and independence of causal mechanisms [3]). Editing the knowledge of LLMs, on the other hand, is highly challenging due to the entanglement of their representations. This can be tackled by continual learning and unlearning methods but it is still an open research problem. At worst, it requires re-training an LLM from scratch. By contrast, our causal graph can be easily kept up to date by a dedicated extraction agent performing atomic updates. The inference agent is decorrelated from the information extraction, allowing it to remain effective without needing re-training, even if the causal relationships in the graph change.
> - This is a great question that is out of the scope of this study, as we aim to tackle it in our future work. However, in Section 7, we provide ideas to address the problem (which we also mention in our response to our second concern). One idea is to use majority voting when merging conflicting documents, since causal relationships that appear in multiple documents are deemed more trustworthy than those that occur only once. Moreover, the LLM extraction agent could be paired with standard causal structure discovery methods to provide additional guarantees.
> - The literature on causal reasoning in LLMs (e.g., [4,5]) has found that LLMs typically cannot discover new causal relationships and are not effective emulators of causal structure discovery algorithms. They mainly rely on domain knowledge. However, they are very robust information retrievers. Inspired by this discrepancy, we take advantage of LLMs to extract known causal relationships, i.e. ones that have been verified and explicitly stated by a reliable human source. This is a much simpler task and we verify that the model indeed recovers the true causal structure (as stated in the data) in Appendix G.
>
> [1] Pearl, J. (2009). Causality. Cambridge university press.
>
> [2] Bareinboim, E., Correa, J. D., Ibeling, D., & Icard, T. (2022). On Pearl’s hierarchy and the foundations of causal inference. In Probabilistic and causal inference: the works of judea pearl (pp. 507-556).
>
> [3] Schölkopf, B., Locatello, F., Bauer, S., Ke, N. R., Kalchbrenner, N., Goyal, A., & Bengio, Y. (2021). Toward causal representation learning. Proceedings of the IEEE, 109(5), 612-634.
>
> [4] Zečević, M., Willig, M., Dhami, D. S., & Kersting, K. (2023). Causal parrots: Large language models may talk causality but are not causal. arXiv preprint arXiv:2308.13067.
>
> [5] Jin, Z., Liu, J., Lyu, Z., Poff, S., Sachan, M., Mihalcea, R., ... & Schölkopf, B. (2023). Can large language models infer causation from correlation?. arXiv preprint arXiv:2306.05836.

---

### Official Review · Reviewer_ksfa · 2025-11-01

**Soundness:** 2
**Presentation:** 3
**Contribution:** 2
**Rating:** 6
**Confidence:** 3

**Summary:**

The paper proposes Causal Cartographer, a two-agent framework for causal reasoning over natural-language sources. The framework consists of (1) CTG-Extract, which performs graph-RAG–assisted causal extraction from news articles to build a large causal knowledge base (CausalWorld), and (2) CTG-Reason, which performs step-by-step, causally constrained inference (including counterfactuals) by conditioning only on parents/children along the graph. The authors also introduce “causal blankets” which, along with a K-matching procedure, enable approximating real-world counterfactuals by matching “worlds” across documents. Empirically, on a 400-query dataset (CausalWorld-CR) derived from news in 2020, CTG-Reason attains accuracy on par with, or better than, a CoT baseline while reducing context and output tokens (up to 70% fewer), with especially large efficiency gains on small models.

**Strengths:**

- The paper argues well for why explicit causal constraints can mitigate spurious correlations and reduce inference cost.
- The two-agent split via decomposition of the task as extraction and reasoning enables each agent to focus specifically on its own task.
- The causal-blanket definition and K-Matching Equivalence theorem formalize when matched worlds yield valid counterfactual targets—useful for this emerging evaluation paradigm.
- The reported token/input reductions and output length shrinkage are substantial while maintaining accuracy.

**Weaknesses:**

- The text corpus utilized is 2020 news with focus on economics. What factors led to this choice? How well does this approach perform in other domains?
- The method leans on SCM framing (DAGs), yet the constructed CausalWorld allows cycles/feedback loops (Fig. 6).
- Causal blankets are defined as fully determining the target (deterministic f). Real news variables are often noisy. Can the theorem and agent be generalized to stochastic blankets?

**Questions:**

1) A small set of bridge nodes routes information across communities. Did you measure how removing a top-k bridge node affects the fraction of nodes still usable for counterfactuals and the success rate of K-matching?

2) You remove queries with ≥50 causal paths and rebalance degree skew. How sensitive are results to the “≥50” threshold, and what happens if you keep hard queries?

3) What max recursion depth or search budget do you set for anticausal inference when parents/children are missing, and how often do queries exceed it?

---

> ### Author Response · Authors · 2025-11-20
>
> We thank Reviewer ksfa for their feedback. Regarding your concerns:
> - As mentioned in Section 4.1, we restricted the timespan of the analysis to avoid potential side effects coming from temporal dependencies as we will tackle them in our future work. We focus on news article sources to obtain high-quality data for robustly extracting stated causal relationships. As illustrated in Figure 4, the information spans multiple domains, ranging from economics to health.
> - We are not sure why this is a limitation. Our framework offers greater expressivity than SCMs, as it supports the representation of cycles and feedback loops. The possible theoretical inconsistencies are resolved by treating the same node at different timesteps as distinct nodes in the DAG during inference when it involves components that form a cycle.
> - Yes, the current version of the theorem is compatible with noise and stochastic processes. We follow the SCM definition of causal processes, which represent stochasticity as random variables. In this theoretical framework, causal processes are deterministic functions of endogenous and exogenous variables, the latter can contain the random processes. We use the same representation in our work and allow stochastic mechanisms.
>
> Regarding your questions:
> - This is an interesting question. We did not initially investigate the impact of removing bridge nodes, but we expect that removing them would heavily impact the distribution of counterfactual queries, as such nodes carry information across domains and participate in the creation of causal blankets. In particular, Figure 17 (b) shows that oil prices are involved in 25% of the counterfactual queries.
> We performed a study where we reconstructed queries from an ablated causal graph where the bridge nodes shown in Figure 17 (b) are removed. We find a new distribution of queries where different bridge nodes have a high occurrence, although less prevalent than in the initial dataset, e.g., the BSE sensex index is involved in 14.82% of the new queries, and the petrol prices are involved in 7.04% of the queries. Removing the main bridge nodes heavily impacts the final distribution but does not have a significant impact on the number of counterfactuals that can be generated in practice. The generation algorithm allows building many possible counterfactuals and we only need to select a random subset to create the final dataset (with balancing). We should note, however, that these new queries are based on an incorrect causal graph because one important causal factor has been omitted.
> - We did not investigate the impact of very long dependencies beyond this threshold, although it could be interesting. However, it is worth noting that they represented a small number of cases and only when building observational queries: indeed, the criteria needed for counterfactual matching are harder to meet as the number of causal paths increases. Causal Blankets are harder to build as the number of variables increases while the number of observations remains constant. So, this case was not encountered in the counterfactual settings. We will make this distinction clearer in the paper.
> - We initially included a recursion depth cutoff for anticausal inference, but found out that it was not necessary because no queries exceeded it. This is intuitively explained by the creation of causal blankets that cover the set of possible causes and restrict the maximum search depth. The blankets can be quite large and allow for a high quantity of nodes inside them, but the natural complexity of blanket creation when many causal paths exist effectively restricts the depth of the queries (see response above) and ensures that queries are tractable. In this setting, anticausal inference is bounded by the blanket and never exceeds any threshold.

---

### Official Review · Reviewer_yTwy · 2025-11-04

**Soundness:** 2
**Presentation:** 2
**Contribution:** 2
**Rating:** 4
**Confidence:** 3

**Summary:**

This paper, “Causal Cartographer: From Mapping to Reasoning over Counterfactual Worlds,” proposes a novel two-agent framework for enabling large language models (LLMs) to extract, organize, and reason with causal knowledge from real-world text. The system, called Causal Cartographer, consists of (1) CTG-Extract, a causal extraction agent based on Graph Retrieval-Augmented Generation (Graph-RAG) that builds a large-scale causal graph (“CausalWorld”) from unstructured text (e.g., 500 economic news articles), and (2) CTG-Reason, a counterfactual reasoning agent that performs stepwise inference under causal constraints. The authors introduce theoretical contributions, notably the concept of causal blankets (a generalization of Markov blankets) and a K-matching algorithm for identifying counterfactual pairs of worlds in text data. Experiments compare CTG-Reason with the chain-of-thought-based CausalCoT method on a new dataset, CausalWorld-CR, derived from real-world causal extractions. Results show comparable or better accuracy and reduced computational cost (up to 70% reduction in inference cost), especially for smaller models like o3-mini and LLaMA-3.1-8B.

**Strengths:**

The paper addresses an important gap between abstract causal reasoning and real-world data extraction. Its proposed combination of causal extraction and counterfactual reasoning within an LLM framework is both ambitious and well-motivated. The introduction of CausalWorld, a large-scale, structured repository of 975 nodes and 1337 causal relations, is an impressive resource that could stimulate further research. The integration of Graph-RAG retrieval ensures grounding in prior causal context during extraction, improving coherence and scalability.

**Weaknesses:**

Despite its strengths, the paper has several limitations that hinder its maturity for a top-tier conference. The evaluation is limited in scope and realism: the CausalWorld-CR dataset is constructed via synthetic matching across news articles rather than ground-truth counterfactual data. This raises concerns about the validity of “real-world” claims and the soundness of the evaluation metric.

**Questions:**

The concept of causal blankets (Section 5.1) should be more carefully distinguished from Pearl’s Markov blankets beyond lineage claims.

---

> ### Author Response · Authors · 2025-11-20
>
> We thank Reviewer yTwy for their feedback. Regarding your concerns:
> - We would like to emphasize that the ground-truth of real-world counterfactuals are generally impossible to obtain due to the fundamental problem of causal inference [1]: indeed in natural settings, only the factual world is observed. For this reason, existing counterfactual reasoning benchmarks like Cladder [2] rely on synthetic constructions. By contrast, our work provides a theoretical framework for identifying a subset of counterfactuals for which the ground-truth is available. We provide theoretical guarantees for our proposed counterfactual matching ensuring its validity. While the assumption of Causal Blanket coverage must be met for counterfactuals to be valid, our work allows exploration of real-world data beyond synthetic settings. It contributes to alleviating the limitations of LLMs for causal reasoning by (1) finding subsets of real-world data for which ground-truth counterfactuals can theoretically be found, and (2) using causal constraints to reduce the inference that an LLM needs to compute from a hard counterfactual ($P(Y|do(X),X’,Y’)$) query to a subset of easier observational questions ($P(Y|X)$).
> - We are not sure to understand what the reviewer means by distinguishing the concept of causal blankets from Pearl’s Markov blankets. To the best of our knowledge, the concept of Markov blanket is independent of Pearl’s theory of causality. If the definition of causal blanket is unclear, we are happy to provide clarifications.
>
> [1] Holland, P. W. (1986). Statistics and causal inference. Journal of the American statistical Association, 81(396), 945-960.
>
> [2] Jin, Z., Chen, Y., Leeb, F., Gresele, L., Kamal, O., Lyu, Z., ... & Sachan, M. CLadder: A benchmark to assess causal reasoning capabilities of language models. arXiv 2023. arXiv preprint arXiv:2312.04350.

---

### Meta-Review · Area_Chair_uJpz · 2026-01-02

**Summary:**

The work proposes Causal Cartographer,  a dual agent framework that enablies large language models to extract, organize, and reason with causal knowledge from real-world text. The framework consists of (1) CTG-Extract, which performs graph-RAG–assisted causal extraction from news articles to build a large causal knowledge base (CausalWorld), and (2) CTG-Reason, which performs step-by-step, causally constrained inference (including counterfactuals) by conditioning only on parents/children along the graph. Another contribution is the CausalWorld-CR data set, consisting of real-world counterfactuals obtained from news sources in 2020.

The paper received 4 reviews, 3 of them leaning on the negative and 1 review leaning on the positive side. The major points of contention from the reviewers were as follows:

* Lack of noise variables in the proposed approach. Although the data set constructed is rooted in the real world news sources, lack of noise variables in the individual structural equations, seems to confine the setting to a more determinsitic one, which in a way defeats the purpose of reasoning in real domains.

* Weak experimental validation. Several reviewers pointed the lack of strong experiments. In causality, this is a tricky question as there are no specific benchmarks for several questions and people do have to rely on synthetic experiments. In that respect, having a new counterfactual data set based on real world news sources is a welcome addition, but based on the description I have seen in the paper as well as the reviews + rebuttals, it seems to be a bit too narrow in scope with respect to getting the message across.

**Reviewer Concerns:**

The authors gave a detailed rebuttal to the points raised in the reviews but unfortunately only 1 of the reviewer engaged with the rebuttal. When I read the text I actually feel that some of the points from the reviewers remained.

For example, the point about the "ground-truth counterfactuals". The authors correctly pointed out that this is the fundamental problem of causal inference but then in their reply confirm tha their work is about "identifying a subset of counterfactuals for which the ground-truth is available". For a specific real world scenario this is an impossibility result and thus the overall claim about real-world scanrios that the work wants to advocate seems a bit self-contradictory.

The point regarding the weak experimental section remains. For example, the authors mention that no query reaches the recursion depth cutoff for anticausal inference. This might not hold in a complex real-world scenario. Also, as I mentioned, the new data set does not seem to add much value to the deprived benchmark condition for causal inference.

**Reviewer Scores:**

I do not see any reviewer increasing their score based on my reading of the paper, the reviews as well as the rebuttal due to the outstanding concerns mentioned in the previous section. Thus, I accept rejection for this version of the work. It is a valuable direction for research and I hope that the authors continue in this direction.

---

### Decision · Program_Chairs · 2026-01-26

Reject